# Regulating Au coverage for the direct oxidation of methane to methanol

Yueshan Xu[1,7], Daoxiong Wu [1,7], Qinghua Zhang [2,7], Peng Rao[1], Peilin Deng [1] ✉, Mangen Tang[1], Jing Li[1], Yingjie Hua[3], Chongtai Wang[3], Shengkui Zhong[4], Chunman Jia[1], Zhongxin Liu[1], Yijun Shen[1], Lin Gu[2,5] ✉, Xinlong Tian [1] ✉ & Quanbing Liu [6] ✉

The direct oxidation of methane to methanol under mild conditions is challenging owing to its inadequate activity and low selectivity. A key objective is improving the selective oxidation of the first carbon-hydrogen bond of methane, while inhibiting the oxidation of the remaining carbon-hydrogen bonds to ensure high yield and selectivity of methanol. Here we design ultrathin $Pd_xAu_y$ nanosheets and revealed a volcano-type relationship between the binding strength of hydroxyl radical on the catalyst surface and catalytic performance using experimental and density functional theory results. Our investigations indicate a trade-off relationship between the reaction-triggering and reaction-conversion steps in the reaction process. The optimized $Pd_3Au_1$ nanosheets exhibits a methanol production rate of 147.8 millimoles per gram of Pd per hour, with a selectivity of 98% at 70 °C, representing one of the most efficient catalysts for the direct oxidation of methane to methanol.

Methane is a promising, clean, and cost-effective feedstock for producing high-value chemicals, such as methanol, which is a versatile energy carrier and platform molecule for the synthesis of important bulk chemicals like olefins and aromatics[1,2]. However, cleaving the first C−H bond in $CH_4$ is difficult because of its high bond energy ($439.3 \, kJ \, mol^{-1}$) and large ionization potential energy (13.0 eV)[3]. In addition, the $CH_3OH$ selectivity is uncontrollable because the remaining C-H bonds can be easily oxidized, leading to the overoxidation of $CH_4$ to carbon dioxide ($CO_2$)[4]. The conventional method for indirect conversion of $CH_4$ to $CH_3OH$ involves the $CH_4$ reforming to syngas ($H_2/CO$) and subsequent synthesis of $CH_3OH$ using syngas as feedstock[5]. However, this energy-intensive method does not satisfy the requirements of green chemistry[6,7]. To address these limitations, the direct oxidation of $CH_4$ to $CH_3OH$ (DOMM) under mild conditions was

proposed[1]. This process is referred to as the "holy grail" reaction, and has been at the forefront of academic and industrial research for many decades[8,9].

Recently, researchers proposed that precious metals can serve as promising catalysts for DOMM because they can effectively reduce the energy barriers and improve the reaction kinetics of C−H bond activation in aqueous media at mild temperatures (<80 °C). To date, the highest performance ($91.8 \, mmol \, g^{-1} \, h^{-1}$) and selectivity (92%) of DOMM have been reported for the class of bimetallic PdAu alloy catalysts[10]. However, previous studies on PdAu alloy mainly focus on zero-dimensional (0D) nanoparticles, which suffer from hard-to-control structure regulations[11,12] and random atom arrangements[13]; Consequently, establishing clear structure-activity relationships of PdAu catalysts and evaluating the roles of Pd and Au atoms are remarkably

[1]School of Marine Science and Engineering, Hainan Provincial Key Lab of Fine Chemistry, School of Chemistry and Chemical Engineering, Hainan University, Haikou 570228, China. [2]Beijing National Laboratory for Condensed Matter Physics, Institute of Physics, Chinese Academy of Sciences, Beijing 100190, China. [3]Key Laboratory of Electrochemical Energy Storage and Energy Conversion of Hainan Province, School of Chemistry and Chemical Engineering, Hainan Normal University, Haikou 571158, China. [4]College of Marine Science & Technology, Hainan Tropical Ocean University, Sanya 572022, China. [5]School of Materials Science and Engineering, Tsinghua University, Beijing 100084, China. [6]Guangzhou Key Laboratory of Clean Transportation Energy Chemistry, Guangdong Provincial Key Laboratory of Plant Resources Biorefinery, School of Chemical Engineering and Light Industry, Guangdong University of Technology, Guangzhou 510006, China. [7]These authors contributed equally: Yueshan Xu, Daoxiong Wu, Qinghua Zhang. ✉e-mail: dengpeilin@hainanu.edu.cn; lingu@mail.tsinghua.edu.cn; tianxl@hainanu.edu.cn; Liuqb@gdut.edu.cn

difficult. In addition, DOMM with $H_2O_2$ as the oxidant is a free-radical process, in which hydroxyl radicals (•OH) triggering the breakage of the first C–H bond is vitally important. However, insufficient attention has been devoted to the formation and/or triggering step of •OH[14]. Ultrathin two-dimensional (2D) nanostructures are superior to 0D nanoparticles in terms of their uniformly exposed facets and an ultrahigh fraction of surface atoms[15,16]. Additionally, the extended surface of 2D nanostructures represents an ideal research platform for regulating the performance of corresponding alloys and exploring the reaction mechanism of DOMM.

Herein, we regulated the coverage of Au atoms on ultrathin $Pd_xAu_y$ nanosheets ($Pd_xAu_y$ NS) using a facile galvanic replacement method and discovered a volcano-type performance–structure relationship between DOMM performance and Au atom coverage. Particularly, the optimized $Pd_3Au_1$ NS achieved a $CH_3OH$ production rate of 147.8 mmol $g^{-1}$ $h^{-1}$ with a high selectivity of 98% at 70 °C. Density functional theory (DFT) calculations suggested that the volcano-type relationship was governed by the energy barriers of the reaction-triggering and reaction-conversion steps on the surface of the $Pd_xAu_y$ NS. Moreover, the strength of the M–O bond measured by using the Integrated Crystal Orbital Hamilton Population (ICOHP) method was used as a promising catalytic descriptor (M–O ICOHP) because it was highly correlated with the energy barrier of the reaction-triggering and reaction-conversion steps. Therefore, the reason for the enhanced DOMM performance on $Pd_xAu_y$ NS was elucidated through the volcano-type relationship between the M–O ICOHP and catalytic performance.

## Results

### Synthesis and characterizations of $Pd_xAu_y$ NS

Following a typical synthesis process, the synthesized hexagonal Pd NS was used as seeds to obtain ultrathin $Pd_xAu_y$ NS with different coverage of Au atoms (details are presented in Supporting Information, Fig. 1a, Supplementary Fig. 1 and Table 1). The atomic ratios of Pd/Au were determined by inductively coupled plasma-optical emission spectrometry (ICP-OES) and denoted $Pd_{33}Au_1$ NS, $Pd_6Au_1$ NS, $Pd_3Au_1$ NS, and $Pd_1Au_1$ NS (Supplementary Fig. 2 and Table 2). The average diameters and lengths of the Pd NS were 60 and 30 nm, respectively (Fig. 1b and Supplementary Fig. 3). The special aberration-corrected scanning transmission electron microscopy (AC-STEM) images confirmed the high crystallinity of the Pd NS at the atomic scale, which had a lattice spacing of 0.224 nm for the face-centered cubic Pd (111) surface (Fig. 1c). The average thickness of each Pd NS was approximately 1.5 nm corresponding to seven layers of Pd atoms (Supplementary Fig. 4), being consistent with the atomic force microscopy results (Supplementary Fig. 5a). The hexagonal morphology of the $Pd_xAu_y$ NS was well preserved after replacing Pd atoms with Au atoms; however, a slight increase in their thickness was observed owing to the larger radius of Au atoms (Supplementary Figs. 5b, 6), and $Pd_1Au_1$ NS was eventually transformed into a Pd@Au core-shell structure (Supplementary Fig. 7). As shown in Fig. 1d, the Au atoms (bright dots) exclusively displaced the Pd atoms instead of simply depositing on the Pd NS surface. The incorporation of such few Au atoms had a negligible effect on the lattice spacing of the crystal face of Pd (111) (Fig. 1e). $Pd_6Au_1$ NS maintained the consummate lattice plane of Pd (111) with a larger lattice spacing (0.232 nm), and more Au single atoms and clusters were dispersed around Pd atoms (Fig. 1f, g). For $Pd_3Au_1$ NS with a lattice spacing of 0.240 nm, Au and Pd atoms (bright and dark dots in Fig. 1h, respectively) were uniformly arranged on the surface. The high-angle annular dark field (HAADF) STEM images and corresponding atomically resolved elemental mapping images showed an ordered atomic arrangement and a regular geometry profile (Fig. 1i). As shown in the X-ray diffraction (XRD) patterns of $Pd_xAu_y$ NS, the diffraction peaks shifted to a lower angle as the coverage of Au atoms increased, except for $Pd_1Au_1$ NS with core-shell structure. This result indicated that the lattice spacing was positively dependent on the Pd/Au ratio, in agreement with the AC-STEM results (Supplementary Fig. 8)[17].

The atomic electronic structure and coordination information of the $Pd_xAu_y$ NS were investigated using X-ray photoelectron

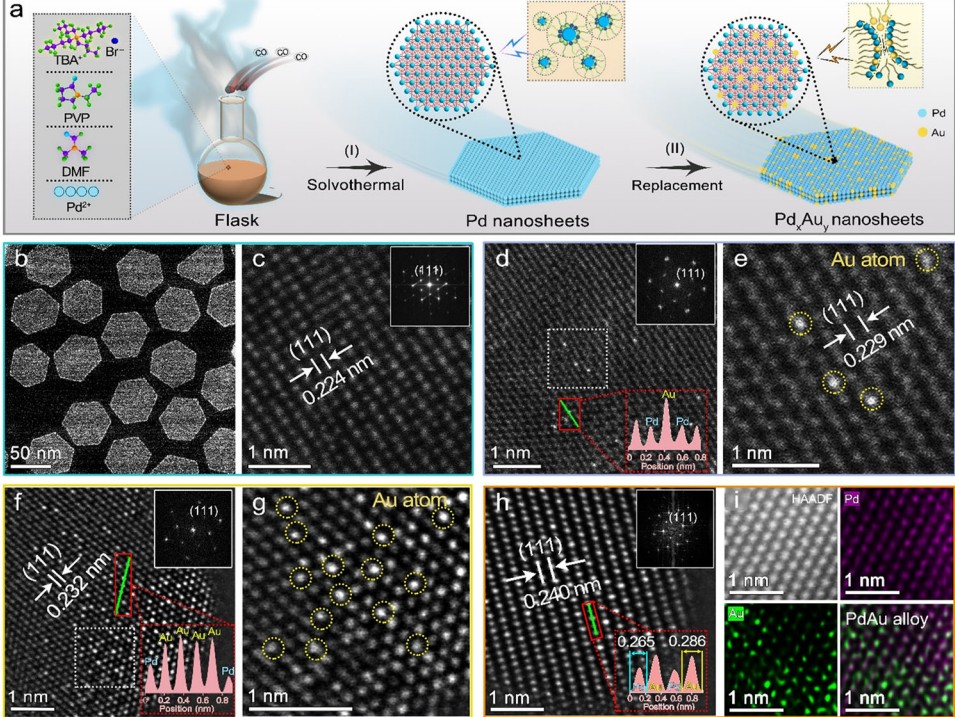

**Fig. 1 | Synthesis and morphology characterization. a** Synthesis schematic of ultrathin $Pd_xAu_y$ NS. **b** Low magnification TEM image of Pd NS. Atomic-resolution AC-STEM images of (**c**) Pd NS, **d, e** $Pd_{33}Au_1$ NS, **f, g** $Pd_6Au_1$ NS, **h** $Pd_3Au_1$ NS. All insert images represent the selected fast Fourier transform images. **i** Atomically resolved elemental mapping images of $Pd_3Au_1$ NS.

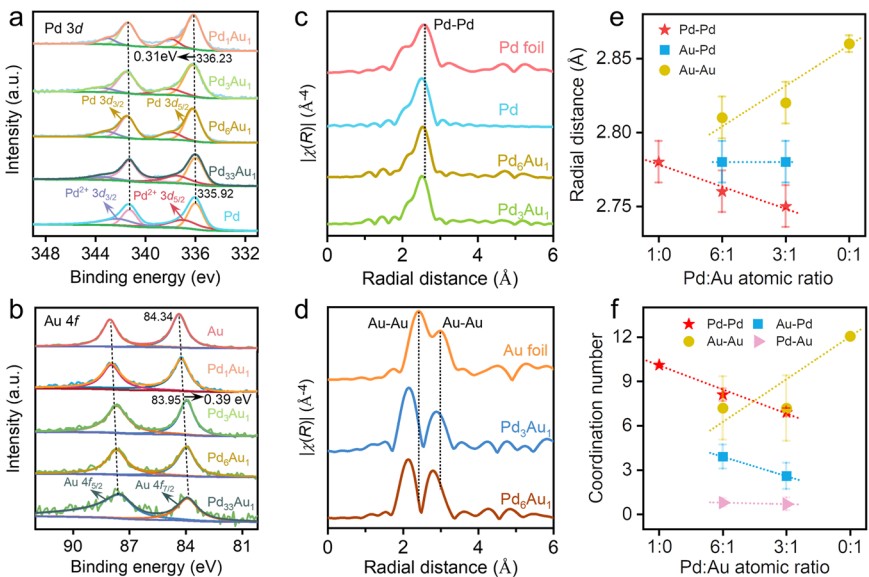

**Fig. 2 | Spectroscopic characterizations. a** XPS spectra of Pd 3*d* for Pd NS and Pd$_x$Au$_y$ NS. **b** XPS spectra of Au 4*f* for Au NS and Pd$_x$Au$_y$ NS. **c, d** The $K^2$-weighted and Fourier-transformed magnitudes of EXAFS spectra of the Pd *K*-edge and Au *L$_3$*- edge. **e** Plots of intermetallic bond length versus Pd/Au ratio. **f** Plots of the coordination number of connected metals versus Pd/Au ratio.

spectroscopy (XPS) and synchrotron X-ray absorption spectroscopy. As shown in Fig. 2a, the peaks of Pd NS at 335.92 and 341.18 eV were attributed to $3d_{5/2}$ and $3d_{3/2}$ of metal Pd$^0$, respectively[18]. In addition, the detected Pd$^{2+}$ species, corresponding to peaks of 336.9 eV ($3d_{5/2}$) and 342.66 eV ($3d_{3/2}$) of Pd NS[19,20] were attributed to slight oxidation of Pd atoms because of the exposure in air. With the increasing Au content, the Pd 3*d* peak of Pd$_x$Au$_y$ NS exhibited a higher binding energy than that of Pd NS, except for the Pd$_1$Au$_1$ NS with the core-shell structure, indicating the electron-deficient state of Pd atoms in Pd$_x$Au$_y$ NS (Supplementary Table 3). For the Au NS, the binding energy of 84.34 and 88.01 eV were attributed to $4f_{7/2}$ and $4f_{5/2}$ of metal Au$^0$ (Fig. 2b). The Au 4*f* peaks of Pd$_x$Au$_y$ NS shifted to the lower binding energy compared with Au NS, and the value increased with increasing the Pd/Au ratio, indicating the significant electronic effects that the electron donation from Pd atoms to Au atoms, which was positively correlated with the coverage of Au atoms (Supplementary Table 4).

In addition, the diffuse reflectance infrared Fourier transform spectroscopy using CO as a probe molecule (CO-DRIFTS) was also performed to assess the electronic state of the Au or Pd of Pd$_x$Au$_y$ NS (Supplementary Fig. 9a). The in-situ CO-DRIFTS spectra showed the physisorption peaks of CO molecule at 2171.7 and 2115.7 cm$^{-1}$ were gradually eliminated over time by sweeping with 10 vol% He/Ar, and the bridged chemisorption peak of CO molecule at 1859.7 and 1915.1 cm$^{-1}$ were present on the surfaces of Pd and Pd$_3$Au$_1$ NS, respectively (Supplementary Figs. 9b, c). Compared with Pd and Au NS, the chemisorption peaks of Pd$_x$Au$_y$ NS exhibited a significant blueshift with increasing Au coverage (Supplementary Fig. 9d)[21–23], indicating the decrease of electron cloud density around Pd atoms, consistent with the XPS results.

The valence state of Pd$_x$Au$_y$ NS was further explored by X-ray absorption near edge structure (XANES) spectroscopy. The energy of the inflection point ($\Delta E$) is the shift in the absorption edge of the sample with respect to that of a standard foil[24]. The higher $\Delta E$ of the Pd *K*-edge signals for Pd$_x$Au$_y$ NS was an indication of slight oxidation of Pd atoms (Supplementary Fig. 10a), whereas the lower valence state of the Au atoms was confirmed by the decreased $\Delta E$ (Supplementary Fig. 10b). The XANES results further confirmed the electronic interactions between Pd and Au atoms in Pd$_x$Au$_y$ NS[24–28]. The atomic coordination information of Pd$_x$Au$_y$ NS near the Pd *K*-edge (Fig. 2c) and

Au *L$_3$*-edge (Fig. 2d) was shown in the $k^4$-weighted and Fourier transform extended X-ray absorption fine structure (EXAFS) spectra. There were no signals indicating the Au–Pd bond, mainly due to the small bond distances of Pd–Pd, Au–Au, and Pd–Au[25,26]. The shifts in both Pd–Pd and Au–Au bonds originate from the strong interactions between Au and Pd atoms, where the Au–Au bond exhibited a considerably higher shift[25]. In addition, the Pd–Pd and Pd–Au bond distances decreased as the coverage of Au atoms increased, whereas the Au–Au bond distance increased gradually due to the assembly of Au atoms (Fig. 2e and Supplementary Fig. 11). Furthermore, as the Au coverage increased, the coordination number of Pd–Pd (CN$_{(Pd-Pd)}$) gradually decreased, whereas the Pd–Au (CN$_{(Pd-Au)}$) increased, suggesting that the Pd atoms were replaced by Au atoms (Fig. 2f and Supplementary Fig. 12). Moreover, the increasing CN$_{(Au-Au)}$ and decreasing CN$_{(Au-Pd)}$, following the increasing coverage of Au atoms, suggested the accumulation of more Au atoms around each Pd atom and the formation of small Au clusters by the assembly of a few Au atoms. Therefore, the observed dependence of the bond distance and coordination number on the coverage of Au atoms demonstrated that the strong electronic interactions were present between Pd and Au atoms in the Pd$_x$Au$_y$ NS (Supplementary Table 5).

## Performance of direct CH$_4$ conversion

The catalytic performance of Pd$_x$Au$_y$ NS was evaluated for DOMM in a pressurized reactor (Supplementary Figs. 13 and 14). Gas chromatography and proton nuclear magnetic resonance ($^1$H NMR) spectroscopy were employed to quantify the gaseous and liquid products (Supplementary Fig. 15). A CH$_3$OH yield of 72.8 mmol g$^{-1}$ h$^{-1}$ with a selectivity of 96% was obtained with Pd NS as the catalyst (Fig. 3a)[29,30]. With increasing the coverage of Au atoms, a volcano-type performance could be observed, and a maximum yield of 147.8 mmol g$^{-1}$ h$^{-1}$ and selectivity of 98% was obtained with the Pd$_3$Au$_1$ NS[4,31]. Although Pd atoms were shown to be the primary active sites for DOMM, the enhanced performance of the Pd$_x$Au$_y$ NS catalyst originated from the Pd atoms affected by catalytically inactive Au atoms (Supplementary Fig. 16 and Supplementary Table 6). Further, the in-situ generation of H$_2$O$_2$ from O$_2$ and H$_2$ rather than the addition of H$_2$O$_2$ or H$_2$O$_2$/O$_2$ as the oxidant was essential for enhancing the performance of DOMM (Supplementary Table 7)[32,33]. The turnover frequencies (TOFs) were

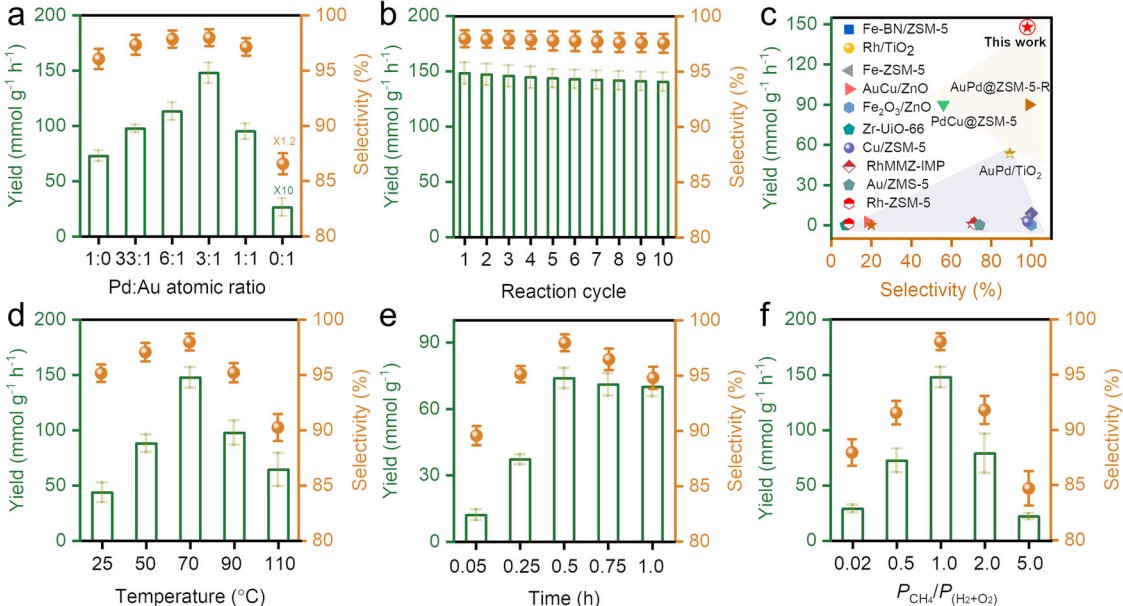

**Fig. 3 | DOMM performance. a** The catalytic performance of DOMM for Pd, $Pd_xAu_y$, and Au NS. **b** Reaction tests for the recycling and regeneration of the $Pd_3Au_1$ catalyst. **c** Comparison of catalytic performance and $CH_3OH$ selectivity for $CH_4$ direct conversion with various catalysts (Supplementary Table 9). **d** The catalytic performance with different reaction temperatures for $Pd_3Au_1$ NS. **e** The catalytic performance with different reaction time at 70 °C for $Pd_3Au_1$ NS. **f** The catalytic performance with different $CH_4$ vol. $Pd_3Au_1$ NS. All other conditions remain the same: 10 mL of water, 1 mg of catalyst, feed gas at 3.0 MPa with 1.1% $H_2$/2.2% $O_2$/ 67.2% $CH_4$/20.57% Ar/8.93 % He. Each reaction was tested three times to obtain the error bars.

determined to evaluate the intrinsic activity of the catalysts, and $Pd_3Au_1$ NS exhibited the highest TOF of all the samples. (Supplementary Table 8). More importantly, the yield and selectivity of $Pd_3Au_1$ NS did not decrease over 10 cycles, suggesting its high-performance stability (Fig. 3b). Furthermore, there was no change in the morphology and structure of $Pd_3Au_1$ NS after the 10-cycle test, confirming its structural stability (Supplementary Fig. 17). As shown in Fig. 3c, the performance of the $Pd_3Au_1$ NS was superior to those of previously reported catalysts under similar normalization conditions (Supplementary Table 9). Additionally, a $CH_3OH$ yield of 43.7 mmol $g^{-1}$ $h^{-1}$ and a selectivity of 95% were obtained for the $Pd_3Au_1$ NS at room temperature, suggesting its considerable potential for industrial application (Supplementary Table 9)[2,34,35]. The effects of various reaction conditions on the $Pd_3Au_1$ NS were systematically investigated. The yield and selectivity of the $CH_3OH$ product had a positive correlation with proper temperature and reaction time. (Fig. 3d, e and Supplementary Tables 10 and 11), and a proper pressure ratio of $CH_4/(H_2 + O_2)$ was considered to be important to regulate the amount of the in-situ generated $H_2O_2$ (Fig. 3f, Supplementary Table 12).

**In situ characterization toward mechanism**

Subsequently, the reaction mechanism of DOMM with the in-situ generation of $H_2O_2$ was analyzed. Time-resolved in-situ diffuse reflectance infrared Fourier transform spectroscopy (DRIFTS) was used to monitor the dynamics of the reaction intermediates on the surface of $Pd_3Au_1$ NS (Fig. 4a)[36,37]. The strong peaks at 3014 and 1303 $cm^{-1}$ were attributed to the antisymmetric stretching $\nu_{as}(C-H)$ and bending $\delta(C-H)$ signal of adsorbed $*CH_4$, respectively (Supplementary Fig. 18)[38]. The peaks at 1450 and 1342 $cm^{-1}$ represented the shear and symmetric shaking vibration of adsorbed $*CH_3$, respectively[39]. The broad peaks appearing in the range of 3200–2700 $cm^{-1}$ and the peak at 1650 $cm^{-1}$ were ascribed to the O−H stretching and bending $\delta(OH)$ signal, respectively[39,40]. In addition, the peak at 1136 $cm^{-1}$ was attributed to the stretching vibration of $*OCH_3$ derived from $CH_3OH$. The small peaks between 2300 and 2400 $cm^{-1}$, assigned to the antisymmetric stretching vibration of the adsorbed $*CO_2$, indicated that the

overoxidation of $CH_4$ to $CO_2$[39]. Upon increasing the reaction time from 0.01 to 1.00 h, both the intensities of the O−H vibration peak and $*CH_3$ peak increased, indicating that the activation of the first C−H bond to form the adsorbed $*CH_3$ species was accomplished by the formation of $*OH$ derived from the dissociation of the in-situ generated of $H_2O_2$. In addition, the signal intensity of $*OCH_3$ gradually increased with increasing the reaction time representing the formation of more $CH_3OH$ product. To investigate the presence of free radicals, electron paramagnetic resonance (EPR) spectroscopy was conducted using 5, 5-dimethyl-1-pyrroline-N-oxide (DMPO) as a spin trap, with the Fenton reaction ($Fe^{2+}$ + $H_2O_2$) for comparison[14,41]. As shown in Fig. 4b, the line (DMPO + $H_2O_2$) represented only the signal peaks of •OH free radical, and the line (DMPO + $CH_3OH$ + $H_2O_2$) suggested the signal peaks of •OH, •OOH, and •$CH_3$ free radicals. Compared with these two lines, the ERP signals of $Pd_3Au_1$ NS presented the coexistence of •OH, •OOH, and •$CH_3$ free radicals during the DOMM process. These results indicated that •OH radical derived from in-situ generated $H_2O_2$ could trigger the activation of C-H bond to form •$CH_3$ radical, and the remaining •OH and •OOH radicals could combine •$CH_3$ radical to form $CH_3OH$ and $CH_3OOH$ products, which was in accordance with DRIFTS results[39]. Subsequently, the relationship between the reaction time and product formation was evaluated using $^1H$ NMR. As shown in Supplementary Fig. 19, $CH_3OOH$ was observed at 3.7 ppm in the early stages[41,42], which was the product of the reaction between •OOH and •$CH_3$ radicals. And $CH_3OH$ was probably obtained from the combination of •$CH_3$ and •OH radicals and the decomposition of $CH_3OOH$.

Temperature-programmed desorption-mass spectrometry (TPD-MS) was used to measure the chemical adsorption strength of the adsorbent species on the catalyst surface[18]. The absence of the desorption peak on Au NS was attributed to the extremely low adsorption capacity for the $CH_4$ molecule (Fig. 4c), while Pd NS exhibited a dominant desorption peak at 386 °C. The lower desorption intensity and temperature of $Pd_3Au_1$ NS implied that a moderate adsorption capacity for the $CH_4$ molecule favored to the DOMM. In addition, thermal programmed desorption−mass spectroscopy (TPD-MS) proved the adsorption of $CH_4$ and $CH_3$ species on $Pd_3Au_1$ NS, which

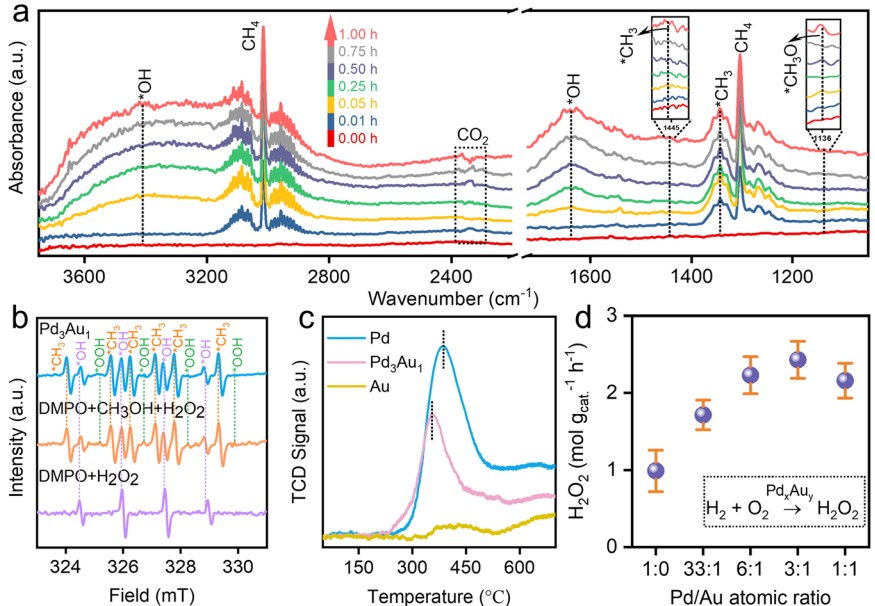

**Fig. 4 | DOMM mechanism. a** In-situ DRIFTS spectra of adsorbed $CH_4$, $O_2$ and $H_2$ at 70 °C for $Pd_3Au_1$ NS in the range of 3700 to 800 $cm^{-1}$. The signal of EPR spectrum of (**b**) radical species (•$CH_3$, •OH, •OOH). **c** $CH_4$-TPD-MS results of Pd, $Pd_3Au_1$ and Au NS. **d** $Pd_xAu_y$ NS-dependency of in-situ generation $H_2O_2$ productivity for 70 °C in $O_2$ and $H_2$ atmosphere. Each reaction was tested three times to obtain the error bars.

suggested that $Pd_3Au_1$ NS could activate the first C–H bond of $CH_4$ without any oxidants at high temperature (Supplementary Fig. 20). The in-situ generation capacity of $H_2O_2$ for $Pd_xAu_y$ NS was examined using titanium oxalate spectrophotometry (Fig. 4d). A clear volcano-type relationship between $H_2O_2$ content and Pd/Au ratio was observed, and $Pd_3Au_1$ NS exhibits the highest $H_2O_2$ production rate of 2.4 mol $g^{-1}$ $h^{-1}$, which is also superior to those of the reported state-of-the-art catalysts (Supplementary Table 13). What's more, the concentration of $H_2O_2$ remained at a high level with increasing reaction time (Supplementary Fig. 21), implying that more free radicals (•OH or •OOH) could remain on the surface of $Pd_3Au_1$ NS. Furthermore, the enhanced capacity of in-situ generation of $H_2O_2$ was explored by $H_2$/$O_2$-TPD measurements. Compared to Pd NS, $Pd_3Au_1$ NS exhibited weaker desorption peaks for $H_2$ and $O_2$ molecules, suggesting that the presence of Au atoms weakened the strong interaction between Pd atom and adsorbed O atom, preventing the O–O band breaking, to form the key intermediate of •OOH (Supplementary Figs. 22 and 23)[43–45].

### Theoretical calculations

DFT calculations were conducted to reveal the origin of the volcano-type structure–performance relationship of $Pd_xAu_y$ NS. Four models with different Pd/Au ratios, namely, pure Pd, $Pd_2Au_1$, $Pd_1Au_2$, and pure Au skin, were adopted (Fig. 5a and Supplementary Fig. 24). As shown in Fig. 5b, efficient DOMM required sufficient •OH radicals to trigger the reaction and a low energy barrier to allow for rapid $CH_4$ conversion to $CH_3OH$[10,11]. Based on the experimental results, the reaction processes of DOMM on the surface of $Pd_xAu_y$ NS involved three key steps, namely, the formation step of the •OH radical (KS-1), activation step of the C–H bond of $CH_4$ (KS-2), and formation step of the $CH_3OH$ product (KS-3), where KS-1 was corresponded to the reaction-triggering step, and KS-2 and KS-3 were corresponded to the reaction-conversion step. The energy barriers of KS-1, KS-2, and KS-3 were denoted $E_{a1}$, $E_{a2}$, and $E_{a3}$, respectively. DFT calculations indicated that the introduction of Au into Pd NS significantly affected the in-situ generation of $H_2O_2$ from $H_2$ and $O_2$ (Supplementary Fig. 25). The monotonic increase in $E_{a1}$ with increasing the coverage of Au atoms confirmed that the introduction of Au could hinder the formation of •OH radicals, which was detrimental to the reaction-triggering step (Fig. 5c and Supplementary

Fig. 26). In addition, the presence of Au atoms did not favor KS-2, as indicated by the larger $E_{a2}$; while it significantly facilitated KS-3, as suggested by the decreased $E_{a3}$ (Supplementary Fig. 27). Combining KS-2 and KS-3, the reaction-conversion step was assessed by the apparent reaction energy barrier ($E_{app}$), defined as the energy difference between the highest energy barrier and initial reaction configuration (Supplementary Fig. 28). The $E_{app}$ exhibited a monotonical decrease from 1.21 to 0.96 eV with increasing the coverage of Au atoms (Fig. 5d and Supplementary Table 14). Among the four models, Pd skin with the lowest $E_{a1}$ exhibited the highest activity for the reaction-triggering step, while Au skin effectively facilitated the reaction-conversion step owing to the lowest $E_{app}$. Therefore, for an effective DOMM process, both the reaction-triggering and reaction-conversion steps needed to be considered, and there was a trade-off between the two steps to achieve optimal performance.

To characterize the aforementioned trade-off effect for the total DOMM process, the reaction rate indicator ($\chi$), defined as $\chi = E_{a1} + E_{app}$, was proposed based on the Arrhenius equation (details are presented in Supplementary Information). Remarkably, when the reaction-triggering and reaction-conversion steps were considered together, a volcano-type relationship between $\chi$ and the coverage of Au atoms could be clearly observed (Fig. 5e), being consistent with the experimental results (Fig. 3a). Therefore, $\chi$ could be used to evaluate the DOMM activity of $Pd_xAu_y$ NS. $Pd_2Au_1$ skin, with the lowest $\chi$ of 1.21 eV, was located at the volcanic peak and estimated to exhibit the best DOMM performance. The performance of the Pd skin (on the left) was limited by the conversion step with a larger $E_{app}$, while the Au-rich surface (on the right) was limited by the triggering step because of the higher $E_{a1}$.

The microscopic mechanisms dominating the volcano-type relationship were further investigated. During the DOMM process, methyl and hydroxyl groups were chemisorbed on the catalyst surface to form M-C and M-O bonds (M represents metal), respectively. From pure Pd skin to pure Au skin, the ability of the $Pd_xAu_y$ NS to adsorb reaction intermediates decreased gradually because Au was more inert than Pd, characterized by the surface binding strength to methyl or hydroxyl groups. The binding strength, evaluated by the ICOHP, exhibited a monotonic decrease from Pd skin to Au skin (Supplementary Fig. 29

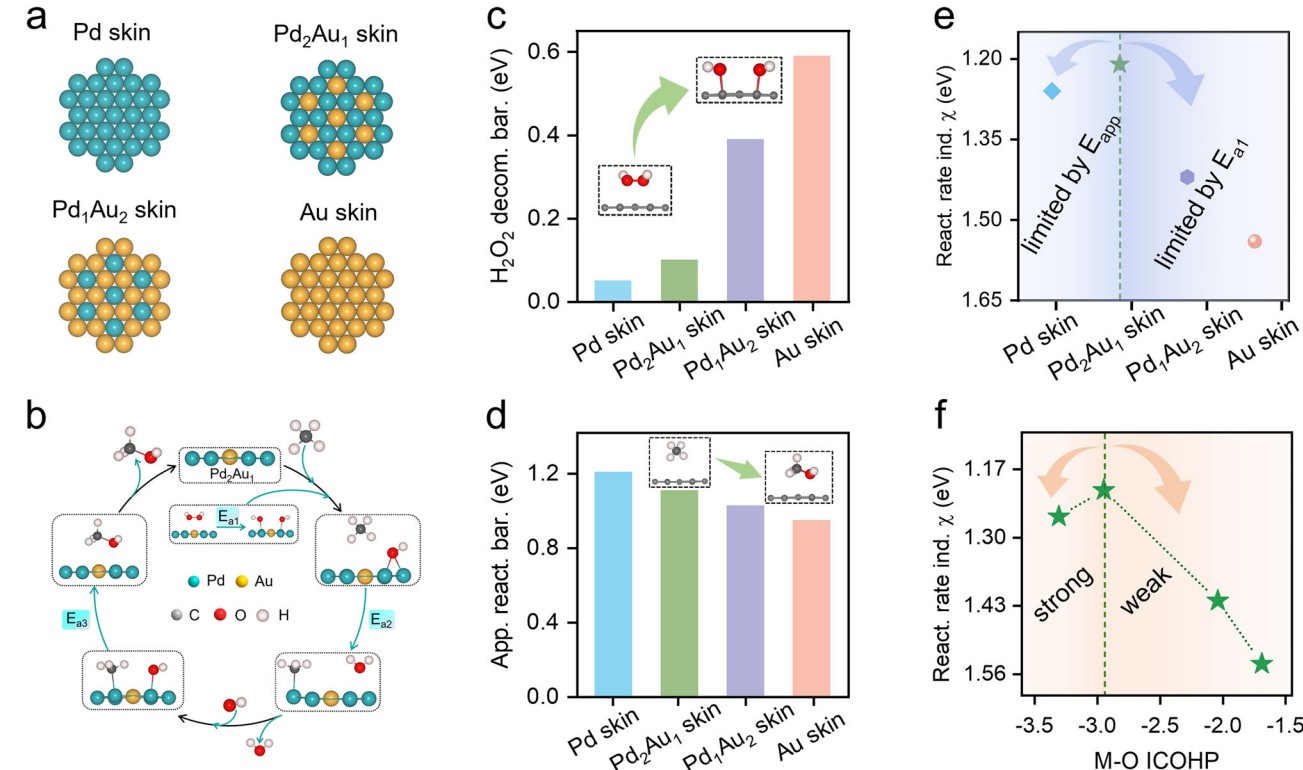

**Fig. 5 | DFT calculations. a** Structural model of four $Pd_xAu_y$ NS. **b** The total reaction pathway of the $CH_4$ oxidation to $CH_3OH$ involves the in-situ generation of $H_2O_2$. The relationship of the coverage of Au atoms dotted Pd surface to **c** the energy barrier of $H_2O_2$ decomposition, **d** apparent reaction energy barrier, and **e** reaction rate indicator χ. **f** The dependence behavior between reaction rate indicator χ and M-O bond strength.

and Table 14). A more negative value of the ICOHP indicated a higher adsorption capacity of the catalyst surface for the reaction intermediates. Given that the •OH radical was both involved in the triggering and conversion steps, and M-O ICOHP was more sensitive to the Pd/Au ratios, thus the M-O ICOHP was applied to analyze the microscopic mechanism of the DOMM. A strong adsorption capacity was needed to facilitate the decomposition of $H_2O_2$ into •OH radicals during the reaction-triggering step (Supplementary Fig. 30a), while a weak adsorption capacity was favorable for the breaking of the M−O bond during the conversion step (Supplementary Fig. 30b). Therefore, a volcano-type relationship between χ and M−O binding strength was established by the trade-off effects (Fig. 5f). Note that this discussion was also applicative to the M-C ICOHP (Supplementary Fig. 31). For catalysts that bind OH groups too strong (M-O ICOHP < −2.95), the performance of DOMM was limited by the slow step of $CH_4$ conversion to $CH_3OH$. By contrast, for catalysts that bind OH groups too weak (M-O ICOHP > −2.95), the performance was limited by insufficient •OH radicals. As DFT calculations suggested, $Pd_2Au_1$ skin with an optimal M-O ICOHP of −2.95 would exhibit the best DOMM performance among the $Pd_xAu_y$ NS. This prediction was confirmed by the experimentally results that $Pd_3Au_1$ NS exhibited the best DOMM performance, with an atomic ratio close to that of $Pd_2Au_1$ skin (see the discussion below the Supplementary Fig. 24). In addition, the $Pd_xAu_y$ skin was thermodynamically unfavorable for further oxidation of •$CH_3$ to methylene species, endowing a high selectivity for $CH_3OH$ (Supplementary Fig. 32).

## Discussion
In this study, we investigated the coverage of Au atoms on $Pd_xAu_y$ NS and established a clear volcano-type structure−performance relationship between $Pd_xAu_y$ NS and their DOMM performance. The maximum yield and selectivity of $Pd_3Au_1$ NS for DOMM were 147.8 mmol $g^{-1}$ $h^{-1}$

and 98.0%, respectively. DFT calculations revealed that two steps, namely, the reaction-triggering and conversion step should be simultaneously considered for an effective DOMM process. The corresponding volcano-type relationship between the DOMM performance and M-O ICOHP suggested that M-O ICOHP could be regarded as a promising catalytic descriptor for this "holy grail" reaction, which could be an effective approach to balance the trade-off effect between the triggering and conversion steps and optimize the performance of $Pd_xAu_y$ catalysts. This study offers not only valuable insights into the reaction mechanisms of DOMM on PdAu alloys, but also a reliable model for developing such alloys with efficient performance.

## Methods
### Materials
Pd(acac)$_2$ (Pd 34.9%, Macklin) and HAuCl$_4$·3H$_2$O (99.9%, Aladdin), Chloro(triphenylphosphine)gold(I) (AuPPh$_3$Cl, 99%, Aladdin), *N, N*-dimethylformamide (DMF, 99.7%, Macklin), Tetrabutylammonium bromide (TBAB, 99%, Aldrich), Potassium titanium oxalate (C$_4$H$_2$K$_2$OTi, 99%, Macklin), 1, 2-dichloropropane (99%, Macklin), 4-tert-butylpyridine (99%, Macklin), Poly(vinylpyrrolidone) (PVP, ~ 29000, Aldrich), Oleylamine (OM, 70%, Aldrich) and *L*-ascorbic acid (AA, 99.5%, Aldrich) and Carbon blacks (xc-72c, Macklin) were used. High-purity water (H$_2$O, 18.3 MΩ cm) was employed for all experiments.

### Preparation of Pd nanosheets
Typically, 185 mg TBAB, 50.0 mg Pd(acac)$_2$, and 160.0 mg PVP were added into 12 mL DMF. The resulting solution was pipetted into a 50 mL glass flask. The flask was then pressurized with CO to 2 bar and kept at 80 °C for 3 h. By centrifugation at 12xg for 1.5 h, the colloidal Pd-blue nanosheets (NS) were precipitated. Finally, the obtained Pd NS were redispersed into 5 mL DMF for further use. The mass yields of the obtained Pd NS were ca. 90−95%.

### Preparation of $Pd_xAu_y$ nanosheets

First, the synthesized Pd nanosheets (NS) and $AuPPh_3Cl$ (3 mg·mL$^{-1}$ in DMF) were premixed into DMF to obtain a certain molar ratio of Pd:Au (Supplementary Table 1). Hydrazine ($N_2H_4\cdot H_2O$, 300 μL, 0.1 mM) was then added drop by drop. Once all the above steps had been completed, the solution was allowed to stand unattended at 25°C for 12 h. The products were collected via centrifuging. Finally, the obtained $Pd_xAu_y$ NS were redispersed into 10 mL $H_2O$ for subsequent applications. The mass yields of the obtained $Pd_xAu_y$ NS were ca. 90–95%.

### Synthesis of Au nanosheets

Typically, 76.2 mL of hexane, 13.2 mL of OM, 1.8 mL of 1, 2-dichloropropane, 0.6 mL of 4-tert-butylpyridine and 30 mL of squalene were well vortexed in a 100 mL glass bottle for 0.08 h. Then this solution was rapidly added to a glass flask with 130 mg $HAuCl_4\cdot 3H_2O$ while continuing to vortex and shake well. The mixed solution was placed in an oven preheated at 58 °C. The product was precipitated after 17 h by centrifugation (5xg, 0.05 h), and then washed three times with hexane and resuspended in hexane. The mass yields of the obtained Au NS were ca. 80–90%.

### Characterization

The TEM and HAADF-STEM mapping images were characterized by three types of TEM instrument (thermoscientific Talos F200X G2; Titan G2 80-200 Chemi-STEM, FEI; and ARM200F, JEOL) operated at 200 kV. X-ray absorption fine structure (XAFS) spectra at Pd $K$-edge and Au $L_3$-edge were performed at BL14W1 station (Shanghai, 3.5 GeV, and 250 mA). XPS was performed using a Shimadzu Axis Supra (Al $K$a and $hv$ = 1486.6 eV). XRD patterns were performed on a Rigaku Smart-Lab operating (Cu $K$a, $\lambda$ = 1.5406 Å, 40 kV, and 40 mA). The Pd and Au loading amounts were determined by the ICP-OES instrument. The tested $CH_3OH$ and $CH_3OOH$ (C1 liquid products) were prepared by adding 300 μL of electrolyte with 250 μL of $D_2O$ and 25 μL of DMSO solution (6 mM). The $^1H$ spectrum peak of DMSO is at -2.6 ppm. The $^1H$ spectrum peak of $D_2O$ is at -4.7 ppm. $^1H$ spectrum peaks of $CH_3OH$ and $CH_3OOH$ are at ~3.3 and ~3.6 ppm, respectively. The total amount ($CH_3OH$ and $CH_3OOH$) was gas chromatography (GC) analyzed, and the amount of $CH_3OOH$ was determined by the minus method. The $CO_2$ was analyzed by GC with FID (Thermo Fisher, T1300). The standard curve method was employed to quantify the content of all products. The following formulae (1) and (2) were employed to calculate the $CH_3OH$ yield and selectivity of all products.

$$CH_3OH \text{ yield}\left(mmol g^{-1}h^{-1}\right) = \frac{CH_3OH (mmol)}{\text{weight of AuPd (g)} \times \text{reaction time (h)}} \quad (1)$$

$$CH_3OH \text{ selectivity}(\%) = \frac{CH_3OH (mmol)}{\text{All products(mmol)}} \times 100\% \quad (2)$$

### Catalytic methane conversion

Methane ($CH_4$) conversion was conducted in a 50 mL autoclave with 1 mg catalyst with 10 wt% $Pd_3Au_1$ supported on carbon blacks. The chamber underwent three times of sealing and flushing using a gas mixture comprising of $H_2/O_2/CH_4/Ar/He$ (3.3%, 6.6%, 1.6%, 61.7%, and 26.8% by volume) and maintained at a pressure of 1.5 MPa. The mixture was agitated at 1.2xg and heated (1.5 °C /min) gradually to a specified temperature (e.g. 70 °C). Then, the autoclave was filled into $CH_4$ gas at the pressure of 3.0 MPa, which continued to keep the target temperature with a regulated reaction time (e.g. 0.5 h). The autoclave was chilled to a temperature below 10 °C in ice in order to minimize the loss of volatile products at the conclusion of the reaction. After each reaction cycle, the catalyst was separated using centrifugation to investigate reusability. The catalyst was used in the next round after drying under vacuum at 80 °C for 12 h.

### Turnover frequency calculation

The Turnover Frequency (TOF) numbers were calculated based following equation:

$$TOF = \frac{n_{CH3OH}}{n_{surface} \times T} = \frac{n_{CH3OH}}{n_{metal} \times T \times \delta} = \frac{n_{CH3OH}}{\frac{m_{cat.} \times w}{M} \times T \times \delta} \quad (3)$$

Where $n_{CH3OH}$ is the amount of substance of $CH_3OH$, $n_{surface}$ is the amount of substance of surface Pd atoms of Pd NS or $Pd_xAu_y$ NS. $n_{metal}$ is the amount of substance of total Pd atoms of Pd NS or $Pd_xAu_y$ NS. $m_{cat.}$ is the mass of the catalyst. $W$ is the mass loading of Pd NS or $Pd_xAu_y$ NS on carbon support, which is measured by using ICP-OES measurement. $M$ is the atomic mass of Pd. $T$ is the reaction time. $\delta$ is the molar percentage of the surface Pd atom of Pd NS or $Pd_xAu_y$ NS.

### Calculation of the molar percentage of surface Pd atom ($\delta$)

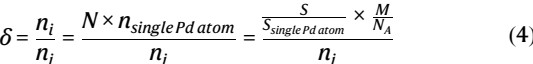

$$\delta = \frac{n_i}{n_j} = \frac{N \times n_{single\,Pd\,atom}}{n_j} = \frac{\frac{S}{S_{single\,Pd\,atom}} \times \frac{M}{N_A}}{n_j} \quad (4)$$

$$S = \frac{3\sqrt{3}}{2} \times t^2 + 6 \times t \times h \quad (5)$$

For one Pd NS, $n_i$ is the amount of substance of surface Pd atom, and $n_j$ is the amount of substance of total Pd atom. $N$ is the number of surface atoms. $S$ is the one Pd NS surface area. $S_{single\,Pd\,atom}$ is single Pd atom surface area. The density, volume, edge length, and thickness are $\rho$, $V$, $t$, and $h$. Where $\rho$ = 12.02 g/cm$^3$, $t$ = 3 × 10$^{-6}$ cm, $h$ = 1.5 × 10$^{-7}$ cm, $V$ = 2.6 × $t^2$ × $h$ = 3.51 × 10$^{-18}$ cm$^3$, $S$ = 2.61 × 10$^{-11}$ cm$^2$, $S_{single\,Pd\,atom}$ = 1.3 × 10$^{-15}$ cm$^2$, $M$ = 106.4 g/mol, $N_A$ = 6.02 × 10$^{-23}$ mol$^{-1}$, $n_j$ = ($\rho$ × $V$)/$M$ = 4.2 × 10$^{-17}$ mol. Thus $\delta_{Pd}$ of Pd NS was calculated as 8.4%. For $\delta_{PdxAuy}$ = $m_{Pdx}$ × $\delta$, $m_{Pdx}$ is the atomic percentage of Pd in $Pd_xAu_y$ NS.

### In-situ diffuse reflectance infrared Fourier transform spectroscopy measurements

In-situ diffuse reflectance infrared Fourier transform spectroscopy (DRIFTS) was used to investigate the dynamic evolution of adsorption species on the catalyst surface. $Pd_xAu_y$ NS were pretreated by heating at 100 °C under Ar flow (20 mL/min) for 1.0 h, and chilled at 70 °C under Ar. Then the gas mixture ($H_2/O_2/CH_4/Ar/He$ = 1.1%/2.2%/67.2%/20.57%/8.93%, v/v) was introduced. DRIFTS spectra were collected with 64 scans of the range 650–4000 cm$^{-1}$ at a resolution of 4 cm$^{-1}$.

In the CO-DRIFTS experiment, the $Pd_xAu_y$ NS was firstly treated in-situ by flowing a 10 vol% He/Ar mixture at 70 °C for 0.5 h. After that, the $Pd_xAu_y$ NS was chilled to 25 °C followed by introducing a 10% CO/Ar mixture at 40 mL/min. Spectra were recorded continuously until the CO adsorption signal reached a constant value. Finally, the gas flow was switched to 10 vol% He/Ar to remove any physically adsorbed species and surface chemisorption spectra were obtained.

### Electron paramagnetic resonance test

Using a Bruker A320 Electron paramagnetic resonance (EPR) spectrometer (1MG (0.1UT)) and 5, 5-dimethyl-1-pyrroline-N-oxide (DMPO) as the scavenger, the free radicals produced during the direct oxidation of $CH_4$ were detected. The DMPO-$H_2O$ solution was mixed with 1 mL of the reaction mixture (100 mmol L$^{-1}$), and immediately transferred to a capillary tube (0.1 mm in diameter with a liquid fill height of approximately 5 cm). The distinctive peaks in the spectrometer's

resonant cavity were then used to identify the type of free radical. For instance, experiment 1 (DMPO + $H_2O_2$ + $Fe^{2+}$) was composed of $FeSO_4 \cdot 7H_2O$ and $HNO_3$ solution (1 mL, 50 mmol $L^{-1}$, and pH = 4) combined with DMPO solution (1 mL, 100 mmol $L^{-1}$).

## Methane thermal programmed desorption test

Samples were gradually heated in a vacuum chamber, and the desorption process occurred when enough energy was available to overcome the desorption activation barrier of the species. After the adsorbates were removed from the catalyst surface in the gas form, they were analyzed by the mass spectrometer. Details are as follows:

The sample was weighed with 50 mg and dried from 25 °C to 150 °C at a rate of 10 °C/min. Ar gas (50 mL/min) was used to purify the sample for 1 h, after which the sample was chilled to 50 °C. Then, 50 mL of $CH_4$/Ar (5%/95%, v/v) mixture was added at the same temperature for 0.5 h. Ar gas stream was replaced for 0.5 h to remove weakly bound $CH_4$ from the surface. The gas was identified by using TCD in an Ar environment up to 700 °C (5 °C/min).

## O$_2$ thermal programmed desorption test

To begin with, 100 mg of catalyst was placed into a reaction tube and heated slowly to 150 °C for the purpose of dry pretreatment, followed by purging with He gas (40 mL/min) for 1 h and cooling to 50 °C. Subsequently, 40 mL of a $O_2$/He (10%/90%, v/v) mixture was introduced to reach saturation for 1 h, and the He stream was then utilized to purge for 1 h to eliminate $O_2$ with weak physical adsorption. Finally, the gas was identified using TCD in a He atmosphere up to 700 °C (10 °C/min).

## H$_2$ thermal programmed desorption test

The reaction tube was weighed with 100 mg of catalyst, which was dried at 150 °C (10 °C/min). After that, He gas (50 mL/min) was purged for 1 h. Following this, the temperature was cooled to 50 °C with a 10% $H_2$/Ar mixture until saturation. Subsequently, the Ar gas stream was introduced with 50 mL/min for 1 h to eliminate any $H_2$ with weak physical adsorption. Lastly, the surface was heated to 700 °C (15 °C/min) in an Ar environment and the desorbed gas was collected using TCD.

## Density functional theory calculation

The VASP was utilized to perform spin-polarized DFT calculations, using the PBE functional and the PAW potential. A 500 eV energy cutoff and $10^{-5}$ eV convergence criterion were chosen for self-consistent calculations. For the purpose of the study, four models were constructed, based on a Pd (111) slab with a supercell ($4 \times 4 \times 1$) and 4 metal layers, including Pd skin, $Pd_2Au_1$ skin, $Pd_1Au_2$ skin, and Au skin. The vacuum layer between adjacent slab models had a thickness of approximately 15 Å and all structures were completely relaxed until the total force on each atom was below 0.05 eV/ Å. For sample the first Brillouin zone, a Γ-centered $k$-point mesh supercell was employed, while the DFT-D3 scheme was utilized to correct for van der Waals interactions. The COHP analysis was executed using the pbeVaspFit2015 base set in the LOBSTER code. The basis functions of $1s$, $2s2p$, $2s2p$, $4d5s$, and $5d6s$ were H, C, O, Pd, and Au, respectively. The CI-NEB method was used to determine the energy barriers. To account for the solvent effect, the implicit solvation model via the VASPsol code. Computations were pre- and post-processed with VASPKIT code and VESTA software.

## Reaction rate indicator ($\chi$)

According to the Arrhenius equation, there was an exponential relationship between the reaction energy barrier ($E_a$) and the reaction rate (R). The exponential relationship could be expressed as R = A×exp[−$E_a$/($k_B$T)], where A, $k_B$, and T were the pre-exponential factors that are considered as a constant, Boltzmann constant, and temperature.

Direct $CH_4$ oxidation to $CH_3OH$ involved the reaction triggering step and reaction conversion step, where the former was governed by $H_2O_2$ decomposition barrier (denoted as $E_{a1}$) and the latter was governed by the apparent reaction barrier (denoted as $E_{app}$). Thus, the total reaction rate of the $CH_4$ conversion to $CH_3OH$ could be described by the product of the Arrhenius relationship of these two processes: $R_{CH4 \to CH3OH}$ = A × exp[−$E_{a1}$/($k_B$T)] × exp[−$E_{app}$/($k_B$T)]. This equation could be simplified as $R_{CH4 \to CH3OH}$ = A×exp[−($E_{a1}$ + $E_{app}$)/($k_B$T)]. The value of $R_{CH4 \to CH3OH}$ increased monotonically as ($E_{a1}$ + $E_{app}$) increased. Hence, we proposed that $\chi$ could regarded as the reaction rate indicator on behalf of ($E_{a1}$ + $E_{app}$).

## Data availability

The data that support the findings of this study are available from the corresponding author upon reasonable request. Source data are provided in this paper. Source data are provided with this paper.

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

## Acknowledgements

This work was supported by the National Natural Science Foundation of China (52164028, 22109035, 52274297, 22202053, 22378074, 52074099, 52164029) to X.L.T, P.L.D, J.L and S.K.Z. X.L.T, P.L.D, J.M.L, J.L, Z.Y.K and D.Q.W acknowledge the Start-up Research Foundation of Hainan University ((KYQD(ZR)-20008, 20082, 20083, 20084, 21124, 21125), the Central Government Guides Local Science and Technology Development Projects (ZY2022HN01), the collaborative Innovation Center of Marine Science and Technology, Hainan University (XTCX2022HYC04, XTCX2022HYC05), the specific research fund of The Innovation Platform for Academicians of Hainan Province (YSPTZX202315). Numerical computations were performed on Hefei advanced computing center. S.K.Z and J.M.L thanks the Natural Science Foundation of Hainan Province (221RC585), Hainan Province Science and Technology Special Fund (ZDYF2022GXJS004, ZDYF2023GXJS006).

## Author contributions

X.T. conceived and designed the experiments. Y.X. and Q.L undertook the materials synthesis, characterization, and performance testing. D.W. contributed to the DFT calculations. Q.Z. and L.G. performed the HAADF-STEM. P.R. assisted with the XANES. P.D. and L.G. co-supervised the experiments. M.T. assisted with the NMR. J.L, Y.H., C.W., S.Z., C.J., Z.L., Y.S., and Q.L assisted with data analysis and paper revision. Y.X., P.D., D.W., and X.T. co-wrote the paper. All the authors discussed the results and commented on the manuscript.

## Competing interests

The authors declare no competing interests.
