## [Peer Review File · Nature Communications]

Regulating Au coverage for the direct oxidation of methane to methanolREVIEWER COMMENTS

Reviewer #1 (Remarks to the Author):

Tian et al. provide an interesting contribution to the field of methane oxidation by using PdAu nanosheets as catalysts. This work builds upon the work of Hutchings et al. and others who have used PdAu nanoparticles for this same reaction. Some of the conclusions that the authors reach are similar to the conclusions that others have reached for PdAu nanoparticles for this same reaction. These similar conclusions are:

1. Varying the Pd:Au ratio changes the rate and selectivity for this reaction. (ACS Catal. 2020, 10, 5115 and Catal. Sci. Technol. 2016, 6, 3410)

2. H₂O₂ that is produced in situ leads to superior selectivity versus H₂O₂ that is added directly to the reactor. (Acc. Chem. Res. 2021, 54, 2614 and <https://doi.org/10.1021/acsorginorgau.3c00001>)

One key question for me is: why were two different Au precursors used (HAuCl₄ and AuPPh₃Cl) used for the PdAu alloys and Au nanosheets? These are quite different precursors and their use should be examined carefully. Is it possible to form the PdAu alloys with HAuCl₄ or do larger Au nanoparticles form rather than the single Au atoms that are observed by the authors? The authors should show results of using HAuCl₄ when making the PdAu alloys. It would not surprise me if the triphenylphosphine (TPP) in the AuPPh₃Cl limits Au aggregation due to the steric bulk of the TPP.

In addition to the different structures that may be formed due to the different precursors, the presence of TPP may affect the catalytic performance. TPP is an electron-donating ligand and could be further modifying the electronic state of the Au atoms (Langmuir 2009, 25, 10548 and Chem. Rev. 2008, 108, 3351.) in addition to the Pd. Under the reactions conditions used by the authors, TPP should still be bound to the Au atoms. I wouldn't expect the TPP to be removed at temperatures of less than 180°C. Here, a comparison with Au atoms that do not have bound ligands would be very important. PdAu alloys which use HAuCl₄ as the Au precursor would not have bound ligands. Also, using an electron-withdrawing ligand such as a thiol bound to Au would provide an interesting comparison to investigate. (Langmuir 1998, 14, 17) Furthermore, a hydrophobic organic ligand environment will disfavor H₂O₂ decomposition versus a hydrophilic environment. (Science 2020, 367, 193)

The authors should provide the values for the peaks in their XPS spectra and provide more discussion of their XPS data.

Another technique to assess the electronic state of the Au or Pd would be to look at chemisorbed CO using DRIFTS. This would indicate how electron rich/poor the Au and Pd are.

It seems that some of the catalysis testing was done with unsupported PdAu alloys and some tests used PdAu alloys supported on XC-72. It has been shown in the literature that the support has an important role in affecting the catalytic performance of PdAu alloys in direct methane oxidation to methanol. Please see ACS Catal. 2018, 8, 2567 and Science 2017, 358, 223. The authors should be more clear on which results used supported versus unsupported PdAu alloys. The comparison in Figure 3C may not be a fair comparison as the catalyst from this work appears to be unsupported and it is compared to only supported catalysts.

Figure 3A should show the monometallic Au sample.

The authors should discuss why the catalyst deactivates at higher temperatures and longer reaction

times and provide evidence for their hypothesis.

Why were different Pd:Au ratios used in the DFT calculations as compared to the experimentally synthesized materials? As an example, the most active and selective Pd:Au ratio of 3:1 was not used in the DFT calculations.

In Table S1, the molar ratios in the left-most column do not match with the ratios in the right-most column.

The authors should be commended on the design of their figures.

While the work is interesting, I feel that it may be too preliminary to publish before the above questions/comments are addressed.

Reviewer #2 (Remarks to the Author):

The work describes the preparation, characterisation and activity/selectivity of PdAu nanosheets for direct oxidation of methane to methanol. The work is clearly explained and conclusions are consistent with the data presented. There is one aspect of the work which warrants detailed attention as this catalyst forms the key results of the paper.

line 140: the easier peroxidation of CH₃OH to CO₂ at higher temp and times (last 2 entries in tables S7 and 8) confirm the conclusions drawn, however, it is not clear why there is a decrease in CO₂ formation at 70 deg or at 0.5 hours. An increasingly linear trend would be expected with increasing temp and time and this is seen for all other data points. The anomolous result at 70 degC and 0.5 h should be explained as this is the key data point used in the paper. For example, has this catalyst been prepared multiple times to check this result using different batches?

Additionally a few comments listed below which should be addressed to the satisfaction of the editor before publication.

Line 23 selectivity to methanol

line 33: the footstone is too strong - perhaps a footstone

line 35: of the CH₃OH product is uncontrollable because the remaining

line 37: methanol is prepared from syn gas via methanol synthesis rather than FT

line 43: catalysts cannot change the position of thermodynamic equilibrium and the statement on line 43 is misleading - the effect of the catalyst should be reworded?

line 45: for DOMM

line 51: catalyst surface

line 62: Integrate Crystal Orbital Hamilton Population method should have upper case first letters

line 273 (SI): should the Pd ratio in Pd₁₃Au₁ be higher, (Table S1)? this seems very different to the 37:1 ratio in the catalyst and 1.25 mg of precursor used

line 87: mapping show an

line 90: the 1:1 ratio catalyst does not seem to follow this trend - please comment

line 370: spectra - not spectrums

line 279 (SI): how was this test result obtained?

line 276 (SI): lower case k in kg - also in 282

table S4: reference the values used for the Pd area to calculate the TOF

line 137: it is not clear why the catalyst being active at room temp gives it considerable potential for industrial application

line 154: generation

line 382: it would be helpful to have inserts for the *CH₃O band as its increase is difficult to see. Also there does not seem to be a corresponding increase in the *CH₃ band at 1427 on the increase of the band at 1342, can the authors please comment?

line 383: give more details of what the lines show in Fig 4B.

line 164, 165: the peaks on the nmr should be identified in Fig S15 in addition to the acetic acid. not just in the experimental

line 169: adsorbates leave the surface

line 182: remain rather than maintain?

line 233 (SI) requires 235: an O₂ molecule....is moderate. 237 because the E_{ad} (O₂)

line 209: require

line 216: should this be Pd₃Au₁?

line 223 upper case for Integrated Crystal....etc

line 31, 37 (SI): use

line 57 (SI): check "minusing"?

lines 88 and 69 (SI): check wording

lines 74-80 (SI): the values used for the number of active sites should be made clear to allow the TOF to be easily calculated from the data in the paper

line 156: edge length a or t?

Reviewer #3 (Remarks to the Author):

This manuscript investigated ultrathin Pd_xAu_y nanosheets for efficient oxidation of methane to methanol and revealed a volcano-type relationship between the binding strength of OH on the catalyst surface and the catalytic performance. However, I cannot support its publication in Nature Communications.

1. My major concern is the relationship between the binding strength of OH on the catalyst surface and the catalytic performance. The physical chemistry of this relationship is unclear. I suggest a comparison for the adsorption and activation of other key species, for example, the adsorption and activation of C-H bond, with that for OH.

2. The strength of M-O bond is calculated using Pd, Pd₂Au₁, Pd₁Au₂ and Au as models. Why did the authors use these models? The authors failed to provide any explanation. In fact, TPD-MS showed that Pd₃Au₁ nanosheet could activate the first C-H bond of CH₄ without any oxidants. But the CH₄ activation barriers of Pd₂Au₁ and Pd₁Au₂ are higher than that of Pd. I was confused by these results.

Other comments include:

1. In the XRD patterns, the (111) plane diffraction peaks of the Pd_xAu_y nanosheet left shifted as the coverage of Au atoms increased. However, that of Pd₁Au₁ shifted to the opposite direction. Can the authors explain this phenomenon?
2. In the XPS Pd 3d spectra of Pd_xAu_y nanosheets, an overlap could be considered between the Pd 3d_{5/2} and the Au 4d_{5/2} components for the bimetallic catalysts.
3. The Pd K-edge and Au L₃-edge XANES spectra are unresolved. The value of ΔE is requested to be labeled in the spectra. Why do the Au-Pd bond distance from Fig.2c and the Pd-Au distance from Fig.2d for the same sample have different distances?
4. In Fig. 3a, the Pd₁Au₁ catalyst showed a reduced yield. But its electronic and geometric structures have not been studied. Can the authors explain the ignorance of this sample?

Dear Reviewers and Editor,

Many thanks for your time and work on our manuscript. Based on your valuable comments and suggestions, we have revised the manuscript carefully by supplementing additional characterizations and discussions. We appreciate your comments, which have greatly enriched the manuscript. Details about the revisions and our responses to the reviewers' comments are provided below in our point-by-point response.

Reviewers' comments:

Reviewer #1:

Tian et al. provide an interesting contribution to the field of methane oxidation by using PdAu nanosheets as catalysts. This work builds upon the work of Hutchings et al. and others who have used PdAu nanoparticles for this same reaction. Some of the conclusions that the authors reach are similar to the conclusions that others have reached for PdAu nanoparticles for this same reaction. These similar conclusions are:

1. Varying the Pd:Au ratio changes the rate and selectivity for this reaction. (*ACS Catal.* 2020, 10, 5115 and *Catal. Sci. Technol.* 2016, 6, 3410).

Response: Thanks for your insightful comments. We strongly agree with you that varying the Pd:Au ratio can change the rate and selectivity for this reaction. The first literature (*ACS Catal.* 2020, 10, 5115) uses multivariate linear regression analysis of 143 samples to quantitative study the effect of reaction condition and catalyst properties in the decomposition of H₂O₂ during methane upgrading. However, the calculation results only showed that higher Au/Pd ratio was beneficial for the rate decrease of H₂O₂ self-decomposition. The second literature (*Catal. Sci. Technol.* 2016, 6, 3410) have showed that the rate of methane oxidation with added H₂O₂ is significantly enhanced by depositing copper together with Au/Pd on the surface of TiO₂. However, the whole process of DOMM was affected by multi-step reactions, including the generation of *OH, the activation of C-H bond, and the formation of CH₃OH. In addition, it is not clear that the synergistic effect between Pd active sites and the around Au atoms for these multi-steps reaction.

In this work, we regulated the **coverage of Au atoms on ultrathin Pd_xAu_y nanosheets (Pd_xAu_y NS)**, and deeply explored the relationship between the structure properties of Pd_xAu_y NS and the reaction mechanisms of DOMM, which remained unclear in the previous reports. The highest yield and selectivity of Pd₃Au₁ NSs for DOMM were **147.8 mmol g⁻¹ h⁻¹** and **98.0%**, respectively, which are much better than the recently reported Pd-based catalysts (*Science* **2017**, 358, 223-227, *Nat. Commun.* **2022** 13, 1375; *Angew. Chem., Int. Ed.* **2022**, 61, e2022041.; *J. Am. Chem. Soc.* **2022**, 144, 740-750; *Angew. Chem., Int. Ed.* **2021**, 60, 5811-5815. etc., Supplementary Table 9). It highlights that both the reaction-triggering and reaction-conversion steps, involved by the binding strength of the OH group, needed to be considered for an effective DOMM process. In addition, the strength of the M–O bond measured using the Integrated Crystal Orbital Hamilton Population (ICOHP) method was used as a promising catalytic descriptor (M–O ICOHP) because it was highly correlated with the energy barrier of the reaction-triggering and conversion steps. Therefore, the innovation in our work lied in the development of Pd_xAu_y catalysts with high activity and selectivity, and the elucidation of the microscopic mechanism behind their volcano-type structure–performance relationship, which offered not only valuable insights into the reaction mechanisms of DOMM on PdAu alloys, but also a reliable model for developing such alloys with efficient performance.

2. H₂O₂ that is produced in situ leads to superior selectivity versus H₂O₂ that is added directly to the reactor. (*Acc. Chem. Res.* 2021, 54, 2614 and <https://doi.org/10.1021/acsorginorgau.3c00001>).

Response: Thanks for your valuable comments. We agree with your comments that the in-situ generation of H₂O₂ on the surface of PdAu is highly favorable for the superior selectivity of the DOMM, which also has been demonstrated in our work. However, the previous reports only explored the consumption efficiency of H₂O₂ to understand the relationship between the H₂O₂ and the performance of the DOMM, which is difficult to explain the relationship between the structural properties of PdAu catalyst and the performance of the DOMM. Therefore, we investigated the effect of the structural properties of PdAu catalyst for the two key steps of the DOMM, including the reaction-triggering and reaction-conversion steps. DFT calculations

indicated that the binding strength of OH group on Pd_xAu_y NS surface is related with these steps, and an obvious volcano-type relationship between the binding strength of OH group on Pd_xAu_y NS surface and DOMM performance was established. Therefore, the integration of theoretical calculations and experimental observations in this work has deepened our understanding of the relationship between radical mechanism and structure-performance in the DOMM, thus providing a basis for accelerating the development of catalysts with higher activity and longer lifetime.

3. One key question for me is: why were two different Au precursors used (HAuCl₄ and AuPPh₃Cl) used for the PdAu alloys and Au nanosheets? These are quite different precursors and their use should be examined carefully. Is it possible to form the PdAu alloys with HAuCl₄ or do larger Au nanoparticles form rather than the single Au atoms that are observed by the authors? The authors should show results of using HAuCl₄ when making the PdAu alloys. It would not surprise me if the triphenylphosphine (TPP) in the AuPPh₃Cl limits Au aggregation due to the steric bulk of the TPP.

Response: Thanks for your professional comments. For the preparation of PdAu alloys nanosheets, AuPPh₃Cl is the preferred Au source due to its stereochemical structure to limit the aggregation of Au atoms. In addition, the Au atoms could uniformly disperse on the Pd nanosheets (Fig. 1 and Fig. R1a). However, when HAuCl₄ is served as the Au source, irregular PdAu alloys nanosheets with large Au nanoparticles around the nanosheet edges were formed (*ACS Materials Lett.* 2021, 3, 996-1002, *J. Colloid Interface Sci.* 2022, 628, 53-63.) (Fig.R1b). However, for the preparation of Au NS, the utilization of HAuCl₄ is more advantageous due to the self-assemble Au³⁺ process with long-chain oleylamine into micelles, facilitating the formation of Au NS (Fig. R2a-c), while the bulky steric hindrance of the AuPPh₃Cl molecule is unpropitious for the formation of Au NS. Moreover, AuPPh₃Cl is not soluble in the reaction mixed solvent (Fig. R2d-e, *J. Am. Chem. Soc.* 2021, 143, 4387-4396)).

Fig. R1 (a) Pd₃Au₁ NS with AuPPh₃Cl as a precursor, (b) Pd₃Au₁ NS with HAuCl₄ as a precursor.

Fig. R2 (a-c) Au NS with HAuCl₄ as a precursor, (d-e) Au NS with AuPPh₃Cl as a precursor.

4. In addition to the different structures that may be formed due to the different precursors, the presence of TPP may affect the catalytic performance. TPP is an electron-donating ligand and could be further modifying the electronic state of the Au atoms (*Langmuir* 2009, 25, 10548 and *Chem. Rev.* 2008, 108, 3351.) in addition to the Pd. Under the reaction conditions used by the authors, TPP should still be bound to the Au atoms. I wouldn't expect the TPP to be removed at temperatures of less than 180°C.

Response: Thanks for your valuable comments. As described in the synthesis process of Pd_xAu_y NS, the highly reducible 0.1 mM N₂H₄·H₂O was added drop by drop in N, N-dimethylformamide (DMF) solution with different concentrations of AuPPh₃Cl to form the Pd_xAu_y NS. Because of the well solubleness of TPP in DMF solution (Fig. R3), the Pd_xAu_y NS was washed by DMF with 5 times under ultrasonic centrifugation treatment to remove the residual TPP on the surface of Pd_xAu_y NS. Compared with the infrared signal of reactants

(AuPPh₃Cl, Pd(acac)₂, TBAB, PVP), no peaks of infrared signal can be observed on Pd_xAu_y NS, confirming the clean surface of Pd_xAu_y NS without residual TPP (Supplementary Fig. 13).

Fig. R3 AuPPh₃Cl dissolved in different solvents. (a) DMF, (b) H₂O and (c) DMF/H₂O(V:V, 1:1).

Revision in the revised manuscript (Page 7)

The catalytic performance of the Pd_xAu_y NS was evaluated for DOMM in a pressurized reactor (Supplementary Fig. 12 and 13).

Revision in the revised supplementary information (Page 21)

Supplementary Fig. 13 FTIR spectra of (a) four reactants and (b) the obtained Pd, Pd_xAu_y, and Au NS.

5. Here, a comparison with Au atoms that do not have bound ligands would be very important. PdAu alloys which use HAuCl₄ as the Au precursor would not have bound ligands. Also, using an electron-withdrawing ligand such as a thiol bound to Au would provide an interesting comparison to investigate. (Langmuir 1998,

14, 17) Furthermore, a hydrophobic organic ligand environment will disfavor H₂O₂ decomposition versus a hydrophilic environment. (Science 2020, 367, 193)

Response: Thanks for your valuable comment. As demonstrated by previous reports (*ACS Materials Lett.* 2021, 3, 996-1002, *J. Colloid Interface Sci.* 2022, 628, 53-63.) and the investigations in our work, the H₂SO₄ as the Au source can result in the formation of Au nanoparticles around the nanosheet edges (Fig. R1b). In addition, the characterization results demonstrated the absence of TPP group on Pd_xAu_y NS when AuPPh₃Cl used as the Au source (Supplementary Fig. 13). To explore the effect of the ligand groups for the DOMM, we have successfully designed three ligand groups on surface of Pd₃Au₁ NS, including Pd₃Au₁-SHC₁₂ with electron-absorbing groups (Fig. R4), Pd₃Au₁-TPP with electron-donating groups (Fig. R5), and Pd₃Au₁ NS without ligand group. The results indicate that the modification of the ligand groups (electron-withdrawal and electron-donating group) have a negative impact on the performance of DOMM (Table R1). This phenomenon may be ascribed for the modification of heteroatomic groups near the active site hindering the chemisorption of CH₄/H₂/O₂.

Fig. R4 XPS spectra of (a) Pd 3d and (b) Au 4f and for Pd₃Au₁ NS and Pd₃Au₁-SHC₁₂. XPS spectra of (c) S 2p for Pd₃Au₁-SHC₁₂. FTIR spectra of (d) Pd₃Au₁-SHC₁₂.

Fig. R5 XPS spectra of (a) Pd 3d and (b) Au 4f and for Pd₃Au₁ NS and Pd₃Au₁-TPP. XPS spectra of (c) P 2p for TPP, Pd₃Au₁ NS and Pd₃Au₁-TPP. FTIR spectra of (d) TPP, Pd₃Au₁ NS and Pd₃Au₁-TPP, respectively.

Table R1. The effect of the ligand groups for Pd₃Au₁ NS supported on carbon blacks as catalyst during the direct oxidation of CH₄.

Entry	Reaction conditions	Amount of product (μmol)		Yield (mmol g ⁻¹ h ⁻¹)	Selectivity (%)
		CH ₃ OH	CO ₂		
1	Pd ₃ Au ₁	7.39	0.15	147.8	98.0
2	Pd ₃ Au ₁ -TPP	1.88	0.07	37.6	96.4
3	Pd ₃ Au ₁ -SR ₁₂	0.81	0.10	16.2	89.0

6. The authors should provide the values for the peaks in their XPS spectra and provide more discussion of their XPS data.

Response: Thanks for your comment. More detailed discussion of the XPS spectra in the revised manuscript.

Revision in the revised manuscript (Page 5)

As shown in Fig. 2a, the peaks of Pd NS at 335.92 and 341.18 eV were attributed to 3d_{5/2} and 3d_{3/2} of metal Pd⁰, respectively.¹⁸ In addition, the detected Pd²⁺ species, corresponding to peaks of 336.9 eV (3d_{5/2}) and 342.66 eV (3d_{3/2}) of Pd NS, respectively,^{19, 20} were attributed to slight oxidation of Pd atoms because of

the exposure in air. With the increasing Au content, the Pd 3*d* peak of Pd_xAu_y NS exhibited a higher binding energy than that of Pd NS, except for the Pd₁Au₁ NS with the core-shell structure, indicating the electron-deficient state of Pd atoms in Pd_xAu_y NS (Supplementary Table 3). For the Au NS, the binding energy of 84.34 and 88.01 eV were respectively attributed to 4*f*_{7/2} and 4*f*_{5/2} of metal Au⁰ (Fig. 2b). The Au 4*f* peaks of Pd_xAu_y NS shifted to the lower binding energy compared with Au NS, and the value increased with increasing of the Pd/Au ratio, indicating the significant electronic effects that the electron of Pd atoms was shifted to Au atoms, which was positively correlated with the coverage of Au atoms (Supplementary Table 4).

Revision in the revised supplementary information (Page 42)

Supplementary Table 3. XPS spectra of Pd 3*d* peak for Pd NS and Pd_xAu_y NS.

Catalysts	B.E. (Pd 3 d _{5/2})/eV	B.E. (Pd ²⁺ 3 d _{5/2})/eV	B.E. (Pd 3 d _{3/2})/eV	B.E. (Pd ²⁺ 3 d _{3/2})/eV
Pd	335.92	336.90	341.18	342.66
Pd ₃₃ Au ₁	335.99	337.33	341.25	342.91
Pd ₆ Au ₁	336.15	337.71	341.43	342.99
Pd ₃ Au ₁	336.23	337.88	341.49	343.19

Revision in the revised supplementary information (Page 43)

Supplementary Table 4. XPS spectra of Au 4*f* peak for Au NS and Pd_xAu_y NS.

Catalysts	B.E. (Au 4 f _{7/2}) / eV	B.E. (Au 4 f _{5/2})/eV
Au	84.34	88.01
Pd ₃ Au ₁	83.95	87.58
Pd ₆ Au ₁	83.92	87.56
Pd ₃₃ Au ₁	83.88	87.52

Revision in the revised manuscript (Page 21)

Fig. 2 | Spectroscopic characterizations. **a** XPS spectra of Pd 3d for Pd, Pd_xAu_y, and Au NS. **b** XPS spectra of Au 4f for Au NS and Pd_xAu_y NS. **c, d** The K^2 -weighted and Fourier-transformed magnitudes of EXAFS spectra of the Pd K -edge and Au L_3 -edge. **e** Plots of intermetallic bond length versus Pd/Au ratio. **f** Plots of the coordination number of connected metals versus Pd/Au ratio.

7. Another technique to assess the electronic state of the Au or Pd would be to look at chemisorbed CO using DRIFTS. This would indicate how electron rich/poor the Au and Pd are.

Response: Thanks for your insightful comments. More detailed discussion of the CO-DRIFTS spectra in the revised manuscript.

Revision in the revised manuscript (Page 6)

In addition, the diffuse reflectance infrared Fourier transform spectroscopy using CO as a probe molecule (CO-DRIFTS) was also performed to assess the electronic state of the Au or Pd of Pd_xAu_y NS (Supplementary Fig. 8a). The in-situ CO-DRIFTS spectra showed the physisorption peaks of CO molecule at 2171.7 and 2115.7 cm^{-1} were gradually eliminated over time by sweeping with 10 vol% He/Ar, and the bridged chemisorption peak of CO molecule at 1859.7 and 1915.1 cm^{-1} were respectively present on the surfaces of

Pd and Pd₃Au₁ NS (Supplementary Figs. 8b-c). Compared with Pd and Au NS, the chemisorption peaks of Pd_xAu_y NS exhibited a significant blueshift with increasing Au coverage (Supplementary Fig. 8d),²¹⁻²³ indicating the decrease of electron cloud density around Pd atoms, consistent with the XPS results.

Revision in the revised supplementary information (Page 16)

Supplementary Fig. 8 (a) The test device photograph of in-situ CO-DRIFTS. The in-situ CO-DRIFTS spectra of (b) Pd NS and (c) Pd₃Au₁ NS with the increasing time. (d) The in-situ CO-DRIFTS spectra of Pd NS, Au NS and Pd_xAu_y NS.

8. It seems that some of the catalysis testing was done with unsupported PdAu alloys and some tests used PdAu alloys supported on C. It has been shown in the literature that the support has an important role in affecting the catalytic performance of PdAu alloys in direct methane oxidation to methanol. Please see ACS Catal. 2018, 8, 2567 and Science 2017, 358, 223. The authors should be more clear on which results used

supported versus unsupported PdAu alloys. The comparison in Figure 3c may not be a fair comparison as the catalyst from this work appears to be unsupported and it is compared to only supported catalysts.

Response: Thanks for your valuable comment. In our work, PdAu alloys supported on carbon blacks were conducted for all catalysis tests, which had been highlighted in the manuscript. We agree with your comment that the support has an important role in affecting the catalytic performance of PdAu alloys for the DOMM. Generally, the interfacial sites at support/metal interface may improve the H₂O₂ decomposition rates to decrease the performance of PdAu alloys (*Science* 2017, 358, 223). In addition, some oxides (CeO₂, ZrO₂, Fe₂O₃, etc.) when used as supports also could bring different catalytic performance due to the strong interaction between metal and the support (*Angew. Chem. Int. Ed.* 2013, 52, 1280-1284; *Angew. Chem. Int. Ed.* 2020, 59, 1216-1219). Accordingly, we have carefully checked the effect of carbon support on the H₂O₂ decomposition and the catalytic performance in this work. Firstly, unsupported PdAu alloys, PdAu alloys supported on carbon black, and pure carbon black, were performed to examine the effect of H₂O₂ decomposition. The results clearly showed that the carbon support has negligible effect for the H₂O₂ decomposition rates (Fig. R6a). In addition, unsupported PdAu alloys and PdAu alloys supported on carbon black, were conducted to explore the effect of the support for the DOMM performance (Fig. R6b). The experimental results proved that the support could play a role in inhibiting agglomeration to agglomeration rove the performance of the DOMM (Fig. R6c-d), but Pd_xAu_y NS with non-support exhibited the phenomenon of the which also been observed by similar literatures (*Catal. Today* 2007, 122, 397-402; *Nano Energy* 2020, 71, 104566). However, Pd_xAu_y NS are tended to agglomeration, while can be highly dispersed on the carbon support, and the performance of the Pd_xAu_y NS/C is much higher than that of the bare Pd_xAu_y NS. Additionally, because Pd_xAu_y alloys in our work were loaded on the carbon black to investigate the performance of DOMM, the comparisons in Fig. 3c were focused on the supported catalysts.

Fig. R6 (a) H₂O₂ degradation as a function of time for pure carbon, Pd₃Au₁ NS, and Pd₃Au₁/C. (b) The catalytic performance of DOMM for Pd₃Au₁ NS and Pd₃Au₁/C. HAADF-STEM images of (c) Pd₃Au₁ NS, and (d) TEM images of Pd₃Au₁ NS/C after DOMM test.

9. Figure 3A should show the monometallic Au sample.

Response: Thanks for your comment. The performance of monometallic Au sample has been added in Fig. 3a in the revised manuscript.

Revision in the revised manuscript (Page 23)

Fig. 3 | DOMM performance. **a** The catalytic performance of DOMM for Pd, Pd_xAu_y, and Au NS. **b** Reaction tests for the recycle and regeneration of the Pd₃Au₁ catalyst. **c** Comparison of catalytic performance and CH₃OH selectivity for CH₄ direct conversion with various catalysts. **d** The catalytic performance with different reaction temperatures for Pd₃Au₁ NS. **e** The catalytic performance with different reaction time for Pd₃Au₁ NS. **f** The catalytic performance with different CH₄ vol. Pd₃Au₁ NS. All other conditions remain the same: 10 mL of water, 1 mg of catalyst, feed gas at 3.0 MPa with 1.1% H₂/2.2% O₂/67.2% CH₄/20.57% Ar/8.93 % He. Each reaction was tested three times to obtain the error bars.

10. The authors should discuss why the catalyst deactivates at higher temperatures and longer reaction times and provide evidence for their hypothesis.

Response: Thanks for your insightful comments. Many literatures have demonstrated that the catalytic performance of the DOMM is highly dependent upon the reaction temperature (*Angew. Chem. Int. Ed.* 2013, 52, 1280-1284; *Angew. Chem. Int. Ed.* 2016, 55, 13441-13445), and Huang et al. further proved the volcano-shape of the selectivity and yield of oxygenates with the increasing of reaction temperature (*Nat. Commun.* 2020, 11, 954). Generally, the decrease of the catalytic performance of the DOMM is directly related with the deactivation of catalyst and the formation of H₂O₂. To evaluate the deactivation of catalyst at higher temperature, the catalytic performance was performed with the same Pd₃Au₁ NS at different temperatures: 70 °C, 110 °C, and afterward 70 °C again. The corresponding yields for CH₃OH product at 70 °C, 110 °C and the second 70 °C were 143.1, 70.5 and 135.5 mmol g⁻¹ h⁻¹, respectively (Fig. R7a). These results showed that the decrease of the catalytic performance at high temperature is not resulted from the deactivation of catalyst, rather than the kinetics effect with a temperature-dependent relationship. Firstly, because H₂O₂ can provide the •OH radical to activity the C-H bond of CH₄ and combine •CH₃ radical to form CH₃OH product, the concentration of H₂O₂ should be important for the catalytic performance (*Science* 2020, 367, 193–197; *Angew. Chem. Int. Ed.* 2022, 61, e202204116). Therefore, the capacity of the in-situ generation of H₂O₂ for Pd_xAu_y NS was examined by using titanium oxalate spectrophotometry. The same volcano-type relationship between

the capacity of in-situ generation of H_2O_2 and the reaction temperature clearly proved that the decrease of the catalytic performance of Pd_3Au_1 NS is due to the low generation rate of H_2O_2 because it's susceptible to peroxidation at the higher temperature (Fig. R7b-c).

As for the reaction time, the experimental results showed that the Pd_3Au_1 NS has a stable catalytic performance from 0.05 h to 0.5 h (Fig. 3e). However, when the reaction time was extended from 0.5 h to 1 h, there was not any change for the CH_3OH yield, while there is an increase trend for CO_2 production (Supplementary Table 11). In addition, the relationship of the in-situ generated yield of H_2O_2 and the reaction time showed a volcano-type change for Pd_3Au_1 NS, and the lower H_2O_2 concentration after 0.5 h may be the reason for the low performance (Supplementary Fig. 20).

Fig. R7 For Pd_3Au_1 NS, (a) The DOMM performance at different temperature. (b) The temperature-dependency of in-situ generated yield of H_2O_2 for Pd_3Au_1 NS. (c) The time-dependency of in-situ generated yield of H_2O_2 for Pd_3Au_1 NS. Reaction conditions: 10 mL of water, 1 mg of Pd_3Au_1 NS catalyst, feed gas at 3.0 MPa with 1.1% H_2 /2.2% O_2 /67.2% CH_4 /20.57% Ar /8.93 % He .

Revision in the revised manuscript (Page 23)

Fig. 3 | DOMM performance. **a** The catalytic performance of DOMM for Pd, Pd_xAu_y, and Au NS. **b** Reaction tests for the recycle and regeneration of the Pd₃Au₁ catalyst. **c** Comparison of catalytic performance and CH₃OH selectivity for CH₄ direct conversion with various catalysts. **d** The catalytic performance with different reaction temperatures for Pd₃Au₁ NS. **e** The catalytic performance with different reaction time for Pd₃Au₁ NS. **f** The catalytic performance with different CH₄ vol. Pd₃Au₁ NS. All other conditions remain the same: 10 mL of water, 1 mg of catalyst, feed gas at 3.0 MPa with 1.1% H₂/2.2% O₂/67.2% CH₄/20.57% Ar/8.93 % He. Each reaction was tested three times to obtain the error bars.

Revision in the revised supplementary information (Page 28)

Supplementary Fig. 20 The time-dependency of in-situ generation H₂O₂ productivity for 70 °C in O₂ and H₂ atmosphere.

Revision in the revised supplementary information (Page 51)

Supplementary Table 11. The catalytic performance of Pd₃Au₁ NS for the direct CH₄ oxidation with different reaction time at 70 °C.

Entry	Time (h)	Amount of product (μmol)			Yield (mmol g ⁻¹)	Selectivity (%)
		CH ₃ OH	CH ₃ OOH	CO ₂		
1	0.05	1.21	0.01	0.13	12.1	89.6
2	0.25	3.72	0.05	0.14	37.2	95.1
3	0.50	7.39	0.00	0.15	73.9	98.0
4	0.75	7.11	0.00	0.26	71.1	96.5
5	1.00	6.87	0.00	0.39	70.0	94.7

11. Why were different Pd:Au ratios used in the DFT calculations as compared to the experimentally synthesized materials? As an example, the most active and selective Pd:Au ratio of 3:1 was not used in the DFT calculations.

Response: Thanks for your professional comment. Detailed discussions related to your comment has been added in the supplementary information.

Revision in the revised supplementary information (Page 31)

Supplementary Fig. 23 Atomic-resolution AC-STEM images of (a) Pd_xAu_y NS. Top view (upper) and side views (lower) of structural model for (b) Pd skin, (c) Pd₂Au₁ skin, (d) Pd₁Au₂ skin, and (e) Au skin.

In the prepared Pd_xAu_y NS samples, the distribution of Au and Pd atoms is not completely homogeneous.

There are a variety of Au-Pd localized skin with different atomic arrangement configurations in the Pd_xAu_y

NS samples, such as the pure Pd skin, the localized skin with one Au surrounded by Pd, the localized skin with one Pd surrounded by Au, and the pure Au skin, see Supplementary Fig. 23. Therefore, it is very difficult to reproduce the atomic structure of the prepared samples in one DFT model. Our DFT calculations aim to qualitatively investigate the behavior and microscopic mechanism of the influence of Au on the catalytic activity of Pd. Considering the computational consumption, the four representative models, including pure Pd skin, Pd₂Au₁ skin, Pd₁Au₂ skin, and pure Au skin, are selected in our calculations. Importantly, based on these four models, the DFT results are qualitatively consistent with the experimental results that the incorporation of a small amount of Au atoms into Pd nanosheets is beneficial for improving the performance of the DOMM. In addition, the DFT calculations show that the Pd₂Au₁ skin model exhibits the highest activity among the four models, which is a potentially active region in the catalyst. It is expected that the surface of the Pd₃Au₁ NS is dominated by the Pd₂Au skin, resulting in the highest catalytic performance.

12. In Table S1, the molar ratios in the left-most column do not match with the ratios in the right-most column.

Response: Thanks for your comment. The molar ratios in the left-most column are the feeding ratio of the reactants (Pd NS and AuPPh₃Cl), and the molar ratios in the right-most column is the atomic molar ratio (Pd/Au) for Pd_xAu_y NS. To eliminate the confusion of two kinds of molar ratios, we re-examined the experimental procedures and change the expression (Supplementary Table 1).

Revision in the revised supplementary information (Page 40)

Supplementary Table 1. The detailed amounts of Pd NS seeds and AuPPh₃Cl to prepare Pd_xAu_y NS with different Pd/Au molar ratios.

Catalysts	m_{AuPPh_3Cl} (mg)	n_{Au} (mol)	V_{PdNS}^{ξ} (mL)	n_{Pd} (mol)	Pd/Au molar ratios
Pd _x Au _y NS	1.25	2.54×10^{-6}	5	9.4×10^{-5}	37:1
	7.50	1.52×10^{-5}			6.2:1
	12.50	2.54×10^{-5}			3.7:1
	25.00	5.08×10^{-5}			1.85:1

ξ : 10 mg Pd NS contained in 5 ml DMF solution.

Reviewer #2:

The work describes the preparation, characterization and activity/selectivity of PdAu nanosheets for direct oxidation of methane to methanol. The work is clearly explained and conclusions are consistent with the data presented.

1. There is one aspect of the work which warrants detailed attention as this catalyst forms the key results of the paper. line 140: the easier peroxidation of CH₃OH to CO₂ at higher temp and times (last 2 entries in tables S7 and 8) confirm the conclusions drawn, however, it is not clear why there is a decrease in CO₂ formation at 70 deg or at 0.5 hours. An increasingly linear trend would be expected with increasing temp and time and this is seen for all other data points. The anomalous result at 70 °C and 0.5 h should be explained as this is the key data point used in the paper. For example, has this catalyst been prepared multiple times to check this result using different batches?

Response: Thanks for your insightful comments. We have carefully retested the catalytic performance with different temperatures, reaction times and different batches, and all the data are averaged by three times. The results demonstrated that the selectivity and yield of DOMM exhibited a volcano-shape trend with increasing the reaction temperature, while the yield of CO₂ increased at higher temperature than 70 °C because the higher reaction temperature favors to the peroxidation of CH₃OH to CO₂ (Supplementary Table 10).

As for the reaction time, the experimental results showed that the Pd₃Au₁ NS has a stable catalytic performance from 0.05 h to 0.5 h (Fig. 3e). However, when the reaction time was extended from 0.5 h to 1 h, there was not any change for the CH₃OH yield, while there is an increase trend for CO₂ production (Supplementary Table 11). In addition, the relationship of the in-situ generated yield of H₂O₂ and the reaction time showed a volcano-type change for Pd₃Au₁ NS, and the lower H₂O₂ concentration after 0.5 h may be the reason for the low performance (Supplementary Fig. 20).

Revision in the revised supplementary information (Page 50)

Supplementary Table 10. The catalytic performance of Pd₃Au₁ NS for the direct CH₄ oxidation with different reaction temperatures.

Entry	Temperature (°C)	Amount of product (μmol)			Yield (mmol g ⁻¹ h ⁻¹)	Selectivity (%)
		CH ₃ OH	CH ₃ OOH	CO ₂		
1	25	2.18	0.005	0.11	43.7	95.0
2	50	4.40	0.00	0.13	88.0	97.1
3	70	7.39	0.00	0.15	147.8	98.0
4	90	4.89	0.00	0.25	97.8	95.1
5	110	3.21	0.00	0.35	64.2	90.2

Revision in the revised supplementary information (Page 51)

Supplementary Table 11. The catalytic performance of Pd₃Au₁ NS for the direct CH₄ oxidation with different reaction time at 70 °C.

Entry	Time (h)	Amount of product (μmol)			Yield (mmol g ⁻¹)	Selectivity (%)
		CH ₃ OH	CH ₃ OOH	CO ₂		
1	0.05	1.21	0.01	0.13	12.1	89.6
2	0.25	3.72	0.05	0.14	37.2	95.1
3	0.50	7.39	0.00	0.15	73.9	98.0
4	0.75	7.11	0.00	0.26	71.1	96.5
5	1.00	6.87	0.00	0.39	70.0	94.7

Additionally, a few comments listed below which should be addressed to the satisfaction of the editor before publication.

1. Line 23 selectivity to methanol

Response: Thanks for your comments. These errors have been corrected in the revised manuscript.

Revision in the revised manuscript (Page 2)

A key objective of DOMM is improving the selective oxidation of the first C–H bond of methane while inhibiting the oxidation of the remaining C–H bonds to ensure high yield and selectivity of methanol.

2. line 33: the footstone is too strong - perhaps a footstone

Response: Thanks for your comments. The word “footstone” has been corrected as “perhaps a footstone”.

Revision in the revised manuscript (Page 3)

Methane (CH_4) is a promising, clean, and cost-effective feedstock for producing high-value chemicals, particularly methanol (CH_3OH), which is perhaps a footstone of the chemical industry.^{1,2}

3. line 35: of the CH_3OH product is uncontrollable because the remaining

Response: Thanks for your comments. These errors have been corrected in the revision.

Revision in the revised manuscript (Page 3)

In addition, the CH_3OH selectivity is uncontrollable because the remaining C-H bonds can be easily oxidized, leading to the overoxidation of CH_4 to carbon dioxide (CO_2).⁴

4. line 37: methanol is prepared from syn gas via methanol synthesis rather than FT

Response: Thanks for your professional comment. The mistake has been corrected.

Revision in the revised manuscript (Page 3)

The conventional method for indirect conversion of CH_4 to CH_3OH involves the CH_4 reforming to syngas (H_2/CO) and subsequent synthesis of CH_3OH using syngas as feedstock.⁵

5. line 43: catalysts cannot change the position of thermodynamic equilibrium and the statement on line 43 is misleading - the effect of the catalyst should be reworded?

Response: Thanks for your valuable comment. The effect of the catalyst has been reworded in the revised manuscript.

Revision in the revised manuscript (Page 3)

Recently, researchers found that precious metals can serve as promising catalysts for DOMM because they can effectively reduce the energy barriers and improve the reaction kinetics of C–H bond activation in aqueous media at mild temperatures (<80 °C).

6. line 45: for DOMM

Response: Thanks for your comment. The mistake has been corrected in the revision.

Revision in the revised manuscript (Page 3)

To date, the highest performance (91.8 mmol g⁻¹ h⁻¹) and selectivity (92%) of DOMM have been reported for the class of bimetallic PdAu alloy catalysts.¹⁰

7. line 51: catalyst surface

Response: Thanks for your comment. The sentence has been rewritten in the revision.

Revision in the revised manuscript (Page 3)

In addition, DOMM with H₂O₂ as the oxidant is a free-radical process, in which hydroxyl radicals (•OH) triggering the breakage of the first C–H bond is vitally important; however, insufficient attention has been devoted to the formation and/or triggering step of •OH.¹⁴

8. line 62: Integrate Crystal Orbital Hamilton Population method should have upper case first letters

Response: Thanks for your comment. Done accordingly.

Revision in the revised manuscript (Page 4)

Moreover, the strength of the M–O bond measured by using the Integrated Crystal Orbital Hamilton Population (ICOHP) method was used as a promising catalytic descriptor (M–O ICOHP) because it was highly correlated with the energy barrier of the reaction-triggering and reaction-conversion steps.

Revision in the revised supplementary information (Page 7)

Crystal Orbital Hamilton Population (COHP)¹¹⁻¹³ analysis as implemented in the LOBSTER code¹⁴ was carried out with the pbeVaspFit2015 basis set¹⁴.

9. line 273 (SI): should the Pd ratio in Pd₁₃Au₁ be higher, (Table S1)? this seems very different to the 37:1 ratio in the catalyst and 1.25 mg of precursor used.

Response: Thanks for your valuable comment. The molar ratios in the left-most column are the feeding ratio of the reactants (Pd NS and AuPPh₃Cl), and the molar ratios in the right-most column is the atomic molar ratio (Pd/Au) for Pd_xAu_y NS. To eliminate the confusion of two kinds of molar ratios, we re-examined the experimental procedures and change the expression (Supplementary Table 1).

Revision in the revised supplementary information (Page 40)

Supplementary Table 1. The detailed amounts of Pd NS seeds and AuPPh₃Cl to prepare Pd_xAu_y NS with different Pd/Au molar ratios.

Catalysts	m_{AuPPh_3Cl} (mg)	n_{Au} (mol)	V_{PdNS}^{ξ} (mL)	n_{Pd} (mol)	Pd/Au molar ratios
Pd _x Au _y NS	1.25	2.54×10^{-6}	5	9.4×10^{-5}	37:1
	7.50	1.52×10^{-5}			6.2:1
	12.50	2.54×10^{-5}			3.7:1
	25.00	5.08×10^{-5}			1.85:1

ξ : 10 mg Pd NS contained in 5 ml DMF solution.

10. line 87: mapping show an

Response: Thanks for your comment. The word “mapping images shown an” has been corrected as “mapping show an”.

Revision in the revised manuscript (Page 5)

The high-angle annular dark field (HAADF) STEM images and corresponding atomically resolved elemental mapping show an ordered atomic arrangement and a regular geometry profile (Fig. 1i).

11. line 90: the 1:1 ratio catalyst does not seem to follow this trend - please comment

Response: Thanks for your insightful comments. Regarding Pd NS as a benchmark, Pd_xAu_y NS gradually transforms into PdAu alloy with increasing the Au content, which is proved by the XRD measurements that the diffraction peak are left-shifted (Supplementary Fig. 7). TEM images also demonstrated the alloyed structure of Pd_xAu_y NS. However, as the mole ratio of Pd/Au reaches 1:1, Pd₁Au₁ NS eventually transforms into a Pd@Au core-shell structure (Supplementary Fig. 6). Therefore, the XRD diffraction peaks of Pd₁Au₁ NS show a different trend towards high angles, which is not consistent with the Pd_xAu_y NS with lower Au content.

Supplementary Fig. 7 (a) XRD patterns of Pd and Pd_xAu_y NS. (b) The magnified image of the region around the crystal face of (111).

Revision in the revised supplementary information (Page 14)

Supplementary Fig. 6 (a) Low magnification TEM image of Pd₁Au₁ NS, (b) TEM images of single Pd₁Au₁ NS and corresponding HRTEM images (c) and FFT pattern (d).

Revision in the revised manuscript (Page 5)

The hexagonal morphology of the Pd_xAu_y NS was well preserved after replacing Pd atoms with Au atoms; however, a slight increase in their thickness was observed owing to the larger radius of Au atoms (Supplementary Figs. 4b, 5), and Pd₁Au₁ NS was eventually transformed into a Pd@Au core-shell structure (Supplementary Fig. 6).

12. line 370: spectra - not spectrums

Response: Thanks for your comments. Done accordingly.

Revision in the revised manuscript (Page 21)

Fig. 2 | Spectroscopic characterizations. **a** XPS spectra of Pd 3d for Pd NS and Pd_xAu_y NS. **b** XPS spectra of Au 4f for Au NS and Pd_xAu_y NS.

13. line 279 (SI): how was this test result obtained?

Response: Thanks for your valuable comment. The C_0 represents the concentration of Pd or Au element in solution for Pd_xAu_y NS with unknown molar ratios of Pd/Au, which were obtained through the external standard method. Firstly, the standard curves were drawn with the known Pd or Au concentration and the corresponding signal values of the inductively coupled plasma optical emission spectroscopy (ICP-OES) (Fig. R8). Next, the quantitative Pd_xAu_y NS dissolved in aqua regia of 10 mL was tested to obtain the signals values of Pd and Au elements by using ICP-OES. Finally, the C_0 was calculated according to the standard curves and the signal values obtained by ICP-OES.

Fig. R8 Standard curves of (a) Pd and (b) Au concentration with the corresponding signal values of the ICP-OES.

14. line 276 (SI): lower case k in kg - also in 282

Response: Thanks for your comments. These errors have been corrected in the revised manuscript.

Revision in the revised supplementary information (Page 41)

Supplementary Table 2. The Pd/Au atomic ratios of Pd_xAu_y NS from the ICP-OES measure.

Catalysts	m_0 (g)	V_0 (mL)	Elements	C_0 (mg/L)	Dilution factor: f	C_1 (mg/L) ^ψ	C_x (mg/kg) ^ξ	Pd/Au molar ratio ^Φ
Pd ₃₃ Au ₁	0.0146	10	Au	0.60	100	60	41095.89	33:1
			Pd	10.77		1077	737575.34	
Pd ₆ Au ₁	0.0297	10	Au	4.97	100	497	167426.60	6:1
			Pd	16.36		1636	550872.05	
Pd ₃ Au ₁	0.0273	10	Au	5.61	100	561	205494.51	3:1
			Pd	9.19		919	336634.07	
Pd ₁ Au ₁	0.0305	10	Au	17.31	100	1731	567868.85	1:1
			Pd	10.07		1007	330131.15	

m_0 is weighing by analytical balance.

V_0 is the volume of the fixed volume after dissolution.

C_0 is the test result.

^ψ: $C_1 = C_0(\text{mg/L}) \times f$.

$$\zeta: C_x(mg/kg) = \frac{C_1(mg/L) \times V_0(mL) \times 10^{-3}}{m(g) \times 10^{-3}}$$

$$\Phi: Pd/Au \text{ molar ratio} = \frac{n_{Pd}(mol)}{n_{Au}(mol)} = \frac{C_{x,Pd}(\frac{g}{kg}) \times m_0(kg) \times 10^{-6}}{M(Pd, \frac{g}{mol})} / \frac{C_{x,Au}(\frac{g}{kg}) \times m_0(kg) \times 10^{-6}}{M(Au, \frac{g}{mol})}$$

15. table S4: reference the values used for the Pd area to calculate the TOF

Response: Thanks for your valuable comment. Our work has been found that Pd atoms are the dominant active sites to realize the DOMM, and Au atoms exhibits extremely poor performance for the DOMM (Supplementary Table 6 and Table R3), which is agree with the reported literatures (*Science* 2020, 367, 193–197; *Nat Catal.* 2022, 5, 45–54). Therefore, the TOF of Pd_xAu_y NS is calculated with Pd atoms as the active sites. The calculation process is as follows.

$$TOF = \frac{\text{Turnover number for } CH_3OH \text{ formation (TON)}}{\text{Number of active site (N)}}; \text{ units is } h^{-1}.$$

$$TON = n(CH_3OH) \times N_A;$$

n is the amount of substance of the CH₃OH product; N_A is 6.02×10^{23} .

$$N = \frac{m \times W}{M} \times N_A$$

N is number of Pd atoms in Pd_xAu_y NS; m is the amount of substance of Pd atoms in Pd_xAu_y NS; M is the relative atomic mass of Pd (106.4 g mol⁻¹); W is the mass fraction of Pd atoms in Pd_xAu_y NS, which is measured by using ICP-OES measurement.

Table R3. The corresponding parameters to calculate the TOF of Pd atoms in Pd_xAu_y NS

Catalysts	m (mol)	M (g mol ⁻¹)	W (%)	TOF (h ⁻¹)
Pd	9.4×10^{-7}			41.2
Pd ₃₃ Au ₁	9.1×10^{-7}			56.8
Pd ₆ Au ₁	8.0×10^{-7}	106.4	10	75.4
Pd ₃ Au ₁	7.0×10^{-7}			111.5
Pd ₁ Au ₁	4.7×10^{-7}			107.5

16. line 137: it is not clear why the catalyst being active at room temp gives it considerable potential for industrial application.

Response: Thanks for your valuable comment. The experimental results have demonstrated that the catalytic activity of the DOMM at room temperature was ascribed for high intrinsic activity of Pd atoms regulated by Au atoms in ultrathin Pd₃Au₁ NS, which could efficiently provide abundant •OH radicals to activate the C-H bond of CH₄. More important, the Pd₃Au₁ NS also decreased the energy barrier of the reaction-triggering and conversion step.

In addition, except for the pivotal catalysts, the required reaction conditions also directly determine the feasibility, yield, and cost of the DOMM reaction in industrial production. The DOMM under mild conditions is as the “holy grail” reaction and has been at the forefront of academic and industrial research for many decades. However, the stable and inert nature of CH₄ molecule makes it difficult to activity and conversion without high temperatures (*Chem. Rev.* 2017, 117, 8497–8520.). PdAu alloy catalysts have been developed for selective oxidation of CH₄ to CH₃OH under mild conditions (≤ 70 °C) using H₂O₂ as oxidant (*Science* 2020, 367, 193-196; *Science* 2017, 358, 299–303). However, the DOMM under room-temperature condition does not need extra energy input to facilitate the reaction occurrence, which is extremely significant yet challenging. In our work, Pd₃Au₁ NS has showed the yield of 43.7 mmol g⁻¹ h⁻¹ and the selectivity of 93.3% for CH₃OH product, which is far exceed the reported literatures (*Science* 2020, 367, 193-196; *Chem* 2018, 4, 1902–1910; *Sci. Adv.* 2020, 6, eaaz9776; *J. Am. Chem. Soc.* 2022, 144, 740-750; *Nano Energy* 2021, 82, 105718; *Cell Rep. Phys. Sci.* 2023, 101277.). Therefore, the DOMM under room-temperature condition with Pd₃Au₁ NS as catalyst would offer the more feasibility and lower cost for industrialization.

17. line 154: generation

Response: Thanks for your comments. These errors have been corrected in the revised manuscript.

Revision in the revised manuscript (Page 8)

Upon increasing the reaction time from 0.01 to 1.00 h, both the intensities of the O–H vibration peak and *CH₃ peak increased, indicating that the activation of the first C–H bond to form the adsorbed *CH₃ species was accomplished by the formation of *OH derived from the dissociation of the in-situ generation of H₂O₂.

18. line 382: it would be helpful to have inserts for the *CH₃O band as its increase is difficult to see. Also there does not seem to be a corresponding increase in the *CH₃ band at 1427 on the increase of the band at 1342, can the authors please comment?

Response: Thanks for your insightful comments. More detailed discussion of the DRIFTS spectra in the revised manuscript.

Revision in the revised manuscript (Page 8)

The peaks at 1450 and 1342 cm⁻¹ represented the shear and symmetric shaking vibration of adsorbed *CH₃, respectively.⁴⁰ The broad peaks appearing in the range of 3200–2700 cm⁻¹ and the peak at 1650 cm⁻¹ were ascribed to the O–H stretching and bending δ(OH) signal, respectively.^{40, 41} In addition, the peak at 1136 cm⁻¹ was attributed to the stretching vibration of *OCH₃ derived from CH₃OH. The small peaks between 2300 and 2400 cm⁻¹, assigned to the antisymmetric stretching vibration of the adsorbed *CO₂, indicated that the overoxidation of CH₄ to CO₂.⁴⁰ Upon increasing the reaction time from 0.01 to 1.00 h, both the intensities of the O–H vibration peak and *CH₃ peak increased, indicating that the activation of the first C–H bond to form the adsorbed *CH₃ species was accomplished by the formation of *OH derived from the dissociation of the in-situ generation of H₂O₂. In addition, the signal intensity of *OCH₃ gradually increased with increasing the reaction time represented the formation of more CH₃OH product.

Revision in the revised manuscript (Page 24)

Fig. 4 | DOMM mechanism. **a** In-situ DRIFTS spectra of adsorbed CH₄, O₂ and H₂ at 70 °C for Pd₃Au₁ NS in the range of 3700 to 800 cm⁻¹. The signal of EPR spectrum of **(b)** radical species (\bullet CH₃, \bullet OH, \bullet OOH). **c** CH₄-TPD-MS results of Pd, Pd₃Au₁ and Au NS. **d** Pd_xAu_y NS-dependency of in-situ generation H₂O₂ productivity for 70 °C in O₂ and H₂ atmosphere.

19. line 383: give more details of what the lines show in Fig 4b.

Response: Thanks for your comments. More detailed information has been provided in the revised manuscript, and we also redraw the Fig. 4b to clearly indicate the meaning of these lines.

Revision in the revised manuscript (Page 9)

As shown in Fig. 4b, the line (DMPO+H₂O₂) represented only the signal peaks of \bullet OH free radical, and the line (DMPO+CH₃OH+H₂O₂) suggested the signal peaks of \bullet OH, \bullet OOH, and \bullet CH₃ free radicals. Compared with these two lines, the ERP signals of Pd₃Au₁ NS presented the coexistence of \bullet OH, \bullet OOH, and \bullet CH₃ free radicals during the DOMM process. These results indicated that \bullet OH radical derived from in-situ generated H₂O₂ could trigger the activation of C-H bond to form \bullet CH₃ radical, and the remaining \bullet OH and \bullet OOH radicals

could combine $\bullet\text{CH}_3$ radical to form CH_3OH and CH_3OOH products, which was in accordance with DRIFTS results.⁴⁰

Revision in the revised manuscript (Page 24)

Fig. 4 | DOMM mechanism. **a** In-situ DRIFTS spectra of adsorbed CH_4 , O_2 and H_2 at 70°C for Pd_3Au_1 NS in the range of 3700 to 800 cm^{-1} . The signal of EPR spectrum of **(b)** radical species ($\bullet\text{CH}_3$, $\bullet\text{OH}$, $\bullet\text{OOH}$). **c** CH_4 -TPD-MS results of Pd, Pd_3Au_1 and Au NS. **d** Pd_xAu_y NS-dependency of in-situ generation H_2O_2 productivity for 70°C in O_2 and H_2 atmosphere.

20. line 164, 165: the peaks on the NMR should be identified in Fig S15 in addition to the acetic acid. not just in the experimental

Response: Thanks for your comments. We would like to kindly point out that there seems to be a misunderstanding for your comment. In this work, the products only contains CH_3OH , CH_3OOH and CO_2 . The “acetic acid” mentioned by reviewer should be CH_3OOH (^1H NMR, 3.7 ppm) instead of acetic acid (CH_3COOH (^1H NMR, 2.1 ppm)), which has been demonstrated by ^1H NMR results (Supplementary Fig. 18).

Many literatures also have reported the CH_3OOH product (*Science*, 2017, 358, 223-227; *Science* 2020, 367, 193-196; *Nat. Commun.* 2020, 11, 954; *J. Am. Chem. Soc.* 2023, 145, 10, 5888–5898).

Revision in the revised supporting information (Page 26)

Supplementary Fig. 18 ^1H NMR spectra of liquid products with the reaction times from 0.01 h to 1 h.

21. line 169: adsorbates leave the surface

Response: Thanks for your comments. These errors have already been corrected in the revised manuscript.

Revision in the revised supporting information (Page 5)

After the adsorbates are removed from the catalyst surface in the gas form, then they are analyzed by the mass spectrometer.

22. line 182: remain rather than maintain?

Response: Thanks for your comments. The word has been corrected.

Revision in the revised manuscript (Page 10)

What's more, the concentration of H₂O₂ remained at a high level with increasing the reaction time (Supplementary Fig. 20), implying that more free radicals (\bullet OH or \bullet OOH) could remain on the surface of Pd₃Au₁ NS.

24. line 233 (SI) requires 235: an O₂ molecule...is moderate. 237 because the E_{ad}(O₂)

Response: Thanks for your comments. The sentence was written.

Revision in the revised supplementary information (Page 32)

Ham et al²⁰ showed that the efficient direct synthesis of H₂O₂ from H₂ and O₂ mediated by PdAu catalysts requires a moderate oxygen adsorption capacity on the catalyst surface. Our calculations show that the adsorbed oxygen molecules on the Pd skin and Pd₂Au₁ skin tend to dissociate into oxygen atoms ($E_{ad}(O_2) - E_{ad}(2O) < 0$ eV), while that on the Au skin is weak with ($E_{ad}(O_2) = -0.16$ eV). The adsorption of an O₂ molecule on Pd₁Au₂ skin is moderate ($E_{ad}(O_2) = -0.38$ eV) and its dissociation on the surface is thermodynamically unfavorable because the $E_{ad}(O_2) - E_{ad}(2O)$ value are positive. Thus, it is speculated that the Pd₁Au₂ moieties play an important role in the direct synthesis of H₂O₂ on the PdAu catalyst surface.

25. line 209: require

Response: Thanks for your comments. The mistake has been corrected.

Revision in the revised manuscript (Page 11)

Therefore, for an effective DOMM process, both the reaction-triggering and reaction-conversion steps needed to be considered, and there was a trade-off between the two steps to achieve the optimal performance.

26. line 216: should this be Pd₃Au₁?

Response: Thanks for your comments. This is the Pd₂Au₁ skin model used in our DFT calculations.

27. line 223 upper case for Integrated Crystal....etc

Response: Thanks for your comments. Done accordingly.

Revision in the revised manuscript (Page 4)

Moreover, the strength of the M–O bond measured by using the Integrated Crystal Orbital Hamilton Population (ICOHP) method was used as a promising catalytic descriptor (M–O ICOHP) because it was highly correlated with the energy barrier of the reaction-triggering and reaction-conversion steps.

28. line 31, 37 (SI): use

Response: Thanks for your comments. These errors have already been corrected in the revised manuscript.

Revision in the revised supplementary information (Page 2)

Preparation of ultrathin Pd nanosheets. Typically, 50.0 mg Pd(acac)₂, 160.0 mg PVP and 185 mg TBAB were mixed with 10 mL DMF and 2 mL water. The resulting homogeneous yellow clear solution was transferred to a 50 mL glass pressure vessel. The vessel was then charged with CO to 2 bar and heated at 80 °C for 3 h before it was cooled to room temperature. The dark blue products were collected by centrifuging at 12000 rpm for 1.5 h. Finally, the obtained Pd nanosheets (NS) were re-dispersed into 5 mL DMF for further use.

Preparation of ultrathin PdAu alloys nanosheets. For a typical synthesis of PdAu nanosheets², Pd nanosheets synthesized above and AuPPh₃Cl (3 mg·mL⁻¹ in 3 mL DMF) were mixed together in DMF to give a molar ratio of Pd:Au of ≈ 3 . Then hydrazine (N₂H₄·H₂O, 300 μ L, 0.1 mM) was added dropwise while the mixture was stirred. After all the above were done, the solution was left undisturbed at room temperature for 12 h. The products were precipitated via centrifuging at 12000 rpm for 0.5 h. Finally, the obtained Pd₃Au₁ NS were re-dispersed into 10 mL H₂O for further use.

29. line 57 (SI): check "minusing"?

Response: Thanks for your comments. The error has been corrected.

Revision in the revised supplementary information (Page 3)

The total amount of CH₃OH and CH₃OOH were analyzed by gas chromatograph, and the amount of CH₃OOH was obtained by minus method.

30. lines 88 and 69 (SI): check wording

Response: Thanks for your comments. The wrongly used words have been corrected.

Revision in the revised manuscript (Page 5)

The high-angle annular dark field (HAADF) STEM images and corresponding atomically resolved elemental mapping showed an ordered atomic arrangement and a regular geometry profile (Fig. 1i).

Revision in the revised supplementary information (Page 4)

Methane oxidation was carried out in a 50 mL stainless-steel autoclave containing a quartz liner vessel with 1 mg of catalyst (10 wt% Pd supported on carbon blacks). The autoclave was sealed and purged three times with feed gas containing 3.3 % H₂, 6.6 % O₂, 1.6 % CH₄, and 61.7 % Ar, and 26.8 % He and maintaining the pressure at 1.5 MPa. The mixture solution was stirred with 1200 rpm and heated to target temperature (e.g. 70 °C) with 1.5 °C/min. Then, the reactor was filled into CH₄ gas with the pressure of 3.0 MPa, which continue to keep the target temperature with the controllable reaction time (e.g. 0.5 h). At the end of reaction, the autoclave was cooled in ice to a temperature below low than 10 °C in order to minimize the loss of volatile products. In order to study the reusability of the catalyst, the colloidal catalyst was separated by centrifugation (12000 rpm/30min) after each reaction run. After drying at 80 °C for 12 h in vacuum, the catalyst was reused in the next run.

31. lines 74-80 (SI): the values used for the number of active sites should be made clear to allow the TOF to be easily calculated from the data in the paper

Response: Thanks for your comments. To better understand the calculation process of the TOF, we have filled out a detailed Supplementary Table 8.

The turnover frequency (TOF) of the as-prepared catalysts

$$TOF = \frac{\text{Turnover number for } CH_3OH \text{ formation (TON)}}{\text{Number of active site (N)}}; \text{ units is } h^{-1}.$$

$$TON = n(CH_3OH) \times N_A;$$

n is the amount of substance of the CH_3OH product; N_A is 6.02×10^{23} .

$$N = \frac{m \times W}{M} \times N_A$$

N is number of Pd atoms in Pd_xAu_y NS; m is the amount of substance of Pd atoms in Pd_xAu_y NS; M is the relative atomic mass of Pd (106.4 g mol^{-1}); W is the mass fraction of Pd atoms in Pd_xAu_y NS, which is measured by using ICP-OES measurement.

Revision in the revised supplementary information (Page 47)

Supplementary Table 8. The turnover frequencies (TOF) of the Pd and Pd_xAu_y NS for DOMM.

Catalyst	n_{CH_3OH} (mol) [#]	TON	m_{Pd} (mol) ^ψ	M (g/mol) ^{&}	W (%) ^ξ	TOF/h ⁻¹
Pd	3.64×10^{-6}	2.2×10^{18}	9.4×10^{-7}		10	41.2
$Pd_{33}Au_1$	4.89×10^{-6}	2.9×10^{18}	9.1×10^{-7}		10	56.8
Pd_6Au_1	5.66×10^{-6}	3.4×10^{18}	8.0×10^{-7}	106.42	10	75.4
Pd_3Au_1	7.39×10^{-6}	4.4×10^{18}	7.0×10^{-7}		10	111.5
Pd_1Au_1	4.75×10^{-6}	2.9×10^{18}	4.7×10^{-7}		10	107.5

32. line 156 (SI): edge length a or t ?

Response: Thanks for your comments. The edge length is t . We have made revisions in the manuscript.

Revision in the revised supplementary information (Page 8)

Calculation of the number of Pd_3Au_1 nanosheets (NS) (N_{NS})

For one Pd₃Au₁ nanosheets (NS), the density, mass, volume, edge length and thickness are ρ , m_{total} , V , t , and h . The number of Pd₃Au₁ nanosheets (N_{NS}) was calculated as

$$N_{NS} = \frac{m_{total}}{\rho \cdot V} = \frac{m_{total}}{\rho \cdot \frac{3\sqrt{3}}{2} t^2 \cdot h} \quad (1)$$

where $\rho=15.67 \text{ g/cm}^3$, $t=3 \times 10^{-6} \text{ cm}$, $h=1.6 \times 10^{-7} \text{ cm}$, $V=\frac{3\sqrt{3}}{2} \cdot t^2 \cdot h=3.74 \times 10^{-18} \text{ cm}^3$, and m_{Total} is the total mass that was weighed in the experiment for $m_{Total}=1 \text{ mg}$ and $N_{NS}=1.70 \times 10^{13}$ in 1 mg total mass.

Calculation of the surface area of Pd₃Au₁ nanosheets (NS) (S_{NS})

For one Pd₃Au₁ nanosheets, the surface area is S_{NS} . The surface area of Pd NS (S_{NS}) was calculated as

$$S_{NS} = \frac{3\sqrt{3}}{2} \cdot t^2 \quad (2)$$

The total surface area of Pd₃Au₁ NS (S_{total}) was calculated as

$$S_{total} = N_{NS} \cdot S_{NS} = N_{NS} \cdot \frac{3\sqrt{3}}{2} \cdot t^2 \quad (3)$$

where $N_{NS}=1.70 \times 10^{13}$ in 1 mg total mass, $S_{NS}=\frac{3\sqrt{3}}{2} \cdot t^2=7.794 \times 10^{-6} \text{ cm}^2$ and $S_{total}=N_{NS} \times S=397.5 \text{ cm}^2$.

Reviewer #3:

This manuscript investigated ultrathin Pd_xAu_y nanosheets for efficient oxidation of methane to methanol and revealed a volcano-type relationship between the binding strength of OH on the catalyst surface and the catalytic performance. However, I cannot support its publication in Nature Communications.

1. My major concern is the relationship between the binding strength of OH on the catalyst surface and the catalytic performance. The physical chemistry of this relationship is unclear. I suggest a comparison for the adsorption and activation of other key species, for example, the adsorption and activation of C-H bond, with that for OH.

Response: Thanks for your valuable comment. Detailed discussions related to your concerns has been added in the revised manuscript and supplementary information.

Revision in the revised manuscript (Page 12)

The microscopic mechanisms dominating the volcano-type relationship were further investigated. During the DOMM process, methyl and hydroxyl groups were chemisorbed on the catalyst surface to form M-C and M-O bonds (M represents metal), respectively. From pure Pd skin to pure Au skin, the ability of the Pd_xAu_y NS to adsorb reaction intermediates decreased gradually because Au was more inert than Pd, characterized by the surface binding strength to methyl or hydroxyl groups. The binding strength, evaluated by the ICOHP, exhibited a monotonic decrease from Pd skin to Au skin (Supplementary Fig. 28 and Table 14). A more negative value of the ICOHP indicated higher adsorption capacity of the catalyst surface for the reaction intermediates. Given that the •OH radical was both involved in the triggering and conversion steps, and M-O ICOHP was more sensitive to the Pd/Au ratios, thus the M-O ICOHP was applied to analyze the microscopic mechanism of the DOMM. A strong adsorption capacity was needed to facilitate the decomposition of H₂O₂ into •OH radicals during the reaction-triggering step (Supplementary Fig. 29a), while a weak adsorption capacity was favorable for the breaking of the M–O bond during the conversion step (Supplementary Fig. 29b). Therefore, a volcano-type relationship between χ and M–O binding strength was established by the trade-off effects (Fig. 5f). Note that this discussion was also applicative to the M-C ICOHP (Supplementary Fig. 30). For catalysts that bind •OH groups too strong (M-O ICOHP < -2.95), the performance of DOMM was limited by the slow step of CH₄ conversion to CH₃OH. By contrast, for catalysts that bind •OH groups too weak (M-O ICOHP > -2.95), the performance was limited by insufficient •OH species. As DFT calculations suggested, Pd₂Au₁ skin with an optimal M-O ICOHP of -2.95 would exhibit the best DOMM performance among the Pd_xAu_y NS. This prediction was confirmed by the experimental results that Pd₃Au₁ NS exhibited the best DOMM performance, with an atomic ratio close to that of Pd₂Au₁ skin (see the discussion below the Supplementary Fig. 23). In addition, the Pd_xAu_y skin was thermodynamically unfavorable for further oxidation of •CH₃ to methylene species, endowing a high selectivity for CH₃OH (Supplementary Fig. 31).

Revision in the revised supplementary information (Page 36)

Supplementary Fig. 28 The value of (a) M-O ICOHP and (b) M-C ICOHP for Pd skin, Pd₂Au₁ skin, Pd₁Au₂ skin, and Au skin. For Pd skin, Pd₂Au₁ skin and Pd₁Au₂ skin, M is Pd element, and for Au-skin, M is Au element. The O and C denoted the oxygen atom in OH and carbon atom in CH₃ species adsorbed on the surface, respectively. The strength of M-O and M-C bond is evaluated by the Integrated Crystal Orbital Hamiltonian Population (ICOHP) at the Fermi level. A more negative ICOHP value corresponds to a stronger bond strength.

Revision in the revised supplementary information (Page 38)

Supplementary Fig. 30 (a) H₂O₂ decomposition barrier, (b) apparent reaction barrier, and (c) reaction rate indicator χ versus M-C ICOHP value.

Revision in the revised supplementary information (Page 54)

Supplementary Table 14. Summary of DFT results. The Underlined data are taken from our previously published work.²¹

	Pd skin	Pd ₂ Au ₁ skin	Pd ₁ Au ₂ skin	Au skin
H ₂ O ₂ decomposition energy barrier (eV)	0.05	0.10	0.39	0.59
CH ₄ activation energy barrier (eV)	0.90	1.11	1.04	0.96
CH ₃ OH formation barrier (eV)	1.42	1.28	1.17	1.05
Apparent reaction barrier (eV)	1.21	1.11	1.03	0.96
Reaction rate indicator χ (eV)	1.26	1.21	1.42	1.55
OH adsorption energy (eV)	-1.16	-0.86	-0.67	-0.31
Adsorption energy of O ₂ (eV)	-0.97	-0.80	-0.38	-0.16
Adsorption energy of dissociated O ₂ (eV)	-2.64	-1.20	0.17	1.06
M-O ICOHP	-3.31	-2.95	-2.04	-1.69
M-C ICOHP	-2.44	-2.42	-2.31	-2.26

2. The strength of M-O bond is calculated using Pd, Pd₂Au₁, Pd₁Au₂ and Au as models. Why did the authors use these models? The authors failed to provide any explanation.

Response: Thanks for your professional comment. Detailed discussions related to your comment has been added in Supplementary Fig. 23 in the supplementary information.

Revision in the revised supplementary information (Page 31)

Supplementary Fig. 23 Atomic-resolution AC-STEM images of (a) Pd_xAu_y NS. Top view (upper) and side views (lower) of structural model for (b) Pd skin, (c) Pd₂Au₁ skin, (d) Pd₁Au₂ skin, and (e) Au skin.

In the prepared Pd_xAu_y samples, the distribution of Au and Pd atoms is not completely homogeneous. There are a variety of Au-Pd localized skin with different atomic arrangement configurations in the Pd_xAu_y samples, such as the pure Pd skin, the localized skin punctuated by single Au atoms (Pd₂Au₁ skin), the localized skin punctuated by single Pd atoms (Pd₁Au₂ skin), and the pure Au skin, see Supplementary Fig. 23. Therefore, it is very difficult to reproduce the atomic structure of the prepared samples in the DFT calculations. Our DFT calculations aim to qualitatively investigate the behavior and microscopic mechanism of the influence of Au on the catalytic activity of Pd. Considering the computational consumption, the four representative models were selected in our calculations. Importantly, based on these four models, the DFT results are qualitatively consistent with the experimental results that the incorporation of a small amount of Au atoms into Pd nanosheets is beneficial for improving the performance of the catalyst. In addition, the DFT calculations show that the Pd₂Au₁ skin model exhibits the highest activity among the four models and is considered to be a potentially active region in the catalyst. It is expected that the surface of the Pd₃Au₁ NS is dominated by the Pd₂Au₁ skin, resulting in the highest catalytic performance.

3. In fact, TPD-MS showed that Pd₃Au₁ nanosheet could activate the first C-H bond of CH₄ without any oxidants. But the CH₄ activation barriers of Pd₂Au₁ and Pd₁Au₂ are higher than that of Pd. I was confused by these results.

Response: Thanks for your insightful comments. TPD-MS results proved that Pd₃Au₁ NS has a lower desorption temperature and adsorption intensity compared to Pd NS or Au NS, which demonstrated that a moderate adsorption capacity was beneficial for the DOMM (Fig. R9a). The subsequent formation of molecular fragments of CH₄ and CH₃ proved that higher reaction temperature can activate the first C-H bond of CH₄ on Pd₃Au₁ catalyst without oxidant (Fig. R9b). Although DFT calculations could qualitatively reveal the reaction mechanism of the DOMM to well match the experimental results at mild conditions (*Nat. Catal.* 2022, 5, 45–54; *Nat. Commun.* 2020, 11, 954), while there is poor consistency between DFT calculations and the experimental results at much higher reaction temperature (>300 °C). In addition, the DFT calculations in

our work were conducted in terms of the coexistence of CH₄ and OH group, while the TPD-MS results showed the activation capacity of C-H bond without the consideration for other species. Therefore, there is not a direct relation between the TPD-MS results and DFT calculations.

Fig. R9 (a) CH₄-TPD results of Pd, Pd₃Au₁ and Au NS. (b) CH₄-TPD-MS results of Pd₃Au₁ NS.

Other comments include:

1. In the XRD patterns, the (111) plane diffraction peaks of the Pd_xAu_y nanosheet left shifted as the coverage of Au atoms increased. However, that of Pd₁Au₁ shifted to the opposite direction. Can the authors explain this phenomenon?

Response: Thanks for your comments. Regarding Pd NS as a benchmark, Pd_xAu_y NS gradually transforms into PdAu alloy with increasing the Au content, which is proved by the XRD measurements that the diffraction peak are left-shifted (Supplementary Fig. 7). TEM images also demonstrated the alloyed structure of Pd_xAu_y NS. However, as the mole ratio of Pd/Au reaches 1:1, Pd₁Au₁ NS eventually transforms into a Pd@Au core-shell structure (Supplementary Fig. 6). Therefore, the XRD diffraction peaks of Pd₁Au₁ NS show a different trend towards high angles, which is not consistent with the Pd_xAu_y NS with lower Au content.

Revision in the revised supplementary information (Page 15)

Supplementary Fig. 7 (a) XRD patterns of Pd and Pd_xAu_y NS. (b) The magnified image of the region around the crystal face of (111).

Revision in the revised supplementary information (Page 14)

Supplementary Fig. 6 (a) Low magnification TEM image of Pd₁Au₁ NS, (b) TEM images of single Pd₁Au₁ NS and corresponding HRTEM images (c) and FFT pattern (d).

Revision in the revised manuscript (Page 5)

The hexagonal morphology of the Pd_xAu_y NS was well preserved after replacing Pd atoms with Au atoms; however, a slight increase in their thickness was observed owing to the larger radius of Au atoms

(Supplementary Figs. 4b, 5), and Pd₁Au₁ NS was eventually transformed into a Pd@Au core-shell structure (Supplementary Fig. 6).

2. In the XPS Pd 3d spectra of Pd_xAu_y nanosheets, an overlap could be considered between the Pd 3d_{5/2} and the Au 4d_{5/2} components for the bimetallic catalysts.

Response: Thanks for your professional comments. To observe the XPS peaks of Pd 3d_{5/2} and Au 4d_{5/2}, we zoomed in on the binding energy range of Pd 3d to 358-331 eV (Fig. R10). However, we did not observe the doublet features of Au 4d (334 eV for Au 4d_{5/2} and 352 eV for Au 4d_{3/2}, *Nat Commun.* 2019, 10, 1428; *ACS Catal.* 2018, 8, 2567-2576.). Therefore, it can be approximated that there is no significant overlap interference between Pd 3d_{5/2} and Au 4d_{5/2}.

Fig. R10 XPS spectra of Pd 3d for Pd NS and Pd_xAu_y NS.

3. The Pd K-edge and Au L₃-edge XANES spectra are unresolved. The value of ΔE is requested to be labeled in the spectra. Why do the Au-Pd bond distance from Fig.2c and the Pd-Au distance from Fig.2d for the same sample have different distances?

Response: Thanks for your valuable comment. The value of ΔE has been added in the revised manuscript (Supplementary Table 5). ΔE represents the inner potential correction to account for the difference in the inner potential between the sample and the reference compound, which is not extracted from the spectra directly. In addition, although the peak position and peak shape in R-space are positively correlated with the range of the Fourier transform, the ranges of the Fourier transform of Pd K-edge and Au L₃-edge are different. Therefore,

different distances between the Pd-Au and Au-Pd peak positions in the Pd *K*-edge and Au *L*₃-edge spectra are measured, which is consistent with the recently reported works (*ACS Nano* 2016, 10, 8645–8659). Moreover, the fitting results of EXAFS demonstrate that the Au-Pd bond distance is similar with the Pd-Au bond.

Revision in the revised supplementary information (Page 44)

Supplementary Table 5. Fitting results of the Pd *K*-edge and Au *L*-edge EXAFS data.

Entry	Catalysts	Edge	Shell	CN	R (Å)	ΔE_0 (eV)	R factor
1	Pd	Pd K	Pd-Pd	10.12	2.78±0.01	4.5	0.001
			Pd-Pd	8.1±0.3	2.76±0.01		
2	Pd ₆ Au ₁	Pd K	Pd-Au	0.8±0.2	2.78±0.01	3.2	0.006
			Au-Au	7.19±1.52	2.81±0.01		
		Au L	Au-Pd	3.90±0.58	2.78±0.01	4.7	0.007
			Pd-Pd	6.9±0.2	2.75±0.01		
3	Pd ₃ Au ₁	Pd K	Pd-Au	0.7±0.3	2.76±0.01	3.3	0.006
			Au-Au	7.18 ± 1.57	2.82±0.01		
		Au L	Au-Pd	2.59 ± 0.63	2.78±0.01	4.1	0.012
4	Au foil	Au L	Au-Au	12	2.86±0.004	4.5	0.002

4. In Fig. 3a, the Pd₁Au₁ catalyst showed a reduced yield. But its electronic and geometric structures have not been studied. Can the authors explain the ignorance of this sample?

Response: Thanks for your insightful comments. The morphology of Pd₁Au₁ NS characterized by spherical-aberration-corrected TEM were clearly showed that the Pd NS were covered by a large amount of Au, resulting in the core-shell structure, which was different with the alloyed Pd_xAu_y NS (Supplementary Fig. 6). In addition, the crystalline and electronic structure of Pd₁Au₁ NS was not in accordance with other Pd_xAu_y NS due to the core-shell structure (Supplementary Fig. 7). In addition, the enhanced performance of the DOMM was ascribed to the modulated electronic effect between Pd atoms and surrounding Au atoms, but not covered by Au atoms. Furthermore, the performance of Pd₁Au₁ catalyst is not good as others. Therefore, the attention was not paid on the Pd₁Au₁ NS.

Revision in the revised manuscript (Page 5)

The hexagonal morphology of the Pd_xAu_y NS was well preserved after replacing Pd atoms with Au atoms; however, a slight increase in their thickness was observed owing to the larger radius of Au atoms (Supplementary Figs. 4b, 5), and Pd₁Au₁ NS was eventually transformed into a Pd@Au core-shell structure (Supplementary Fig. 6).

Revision in the revised manuscript (Page 23)

Fig. 3 | DOMM performance. **a** The catalytic performance of DOMM for Pd, Pd_xAu_y, and Au NS. **b** Reaction tests for the recycle and regeneration of the Pd₃Au₁ catalyst. **c** Comparison of catalytic performance and CH₃OH selectivity for CH₄ direct conversion with various catalysts. **d** The catalytic performance with different reaction temperatures for Pd₃Au₁ NS. **e** The catalytic performance with different reaction time for Pd₃Au₁ NS. **f** The catalytic performance with different CH₄ vol. Pd₃Au₁ NS. All other conditions remain the same: 10 mL of water, 1 mg of catalyst, feed gas at 3.0 MPa with 1.1% H₂/2.2% O₂/67.2% CH₄/20.57% Ar/8.93 % He. Each reaction was tested three times to obtain the error bars.

Revision in the revised supplementary information (Page 14)

Supplementary Fig. 6 (a) Low magnification TEM image of Pd₁Au₁ NS, (b) TEM images of single Pd₁Au₁ NS and corresponding HRTEM images (c) and FFT pattern (d).

Revision in the revised supplementary information (Page 15)

Supplementary Fig. 7 (a) XRD patterns of Pd and Pd_xAu_y NS. (b) The magnified image of the region around the crystal face of (111).

Finally, we again appreciate the editor and the reviewers for your valuable time and insightful comments, which were extremely helpful in improving the quality of our paper and enhancing our future work. We also hope that Reviewers will appreciate our great efforts put in the revision, and this version will be deemed acceptable for publication in *Nature Communications*.

REVIEWER COMMENTS

Reviewer #1 (Remarks to the Author):

I have a few questions/edits before recommending for final publication.

1. What is a footstone?
2. Does the catalyst deactivate after 30 minutes?
3. In Figure 3E, there is no temperature listed.

The authors should be commended on their edits since the original manuscript.

Reviewer #2 (Remarks to the Author):

The comments have been mainly addressed, however, a few are still unclear.

Point 2: it was recommended to reword footstone however I see that "perhaps footstone" does not make sense. It would be better to remove the word "perhaps" from the text. This is a personal phrasing and the authors should word as they see best.

point 10: "mapping..." was not the correction rather "shown" changed to "show" which has been done. mapping "images" should be reinserted

Point 13: The query was concerned with adding the details of the ICP method which I am not sure is included in the experimental section

Point 15: The query was asking for the values of the Pd area to be included so allow the calculation of the TOF to be followed. I am still unsure where the areas of Pd are shown and how these surface areas were obtained. See also point 31 below

Point 17: should be "of the in-situ generated H₂O₂"

Point 31: It is still unclear how the number of active sites has been determined. The amount of Pd atoms is relevant for productivity but TOF conventionally used surface sites, please comment on the number of Pd sites which are on the surface. This relates to point 15 above

Reviewer #3 (Remarks to the Author):

The authors have added some new discussion in this revised manuscript to address my comments. However, I still have concerns for the reaction kinetics and the structure characterization of AuPd nanosheets.

1. The authors claimed the existence of a volcano-type relationship between M-O binding strength and catalytic performance. "During the DOMM process, methyl and hydroxyl groups were chemisorbed on the catalyst surface to form M-C and M-O bonds (M represents metal), respectively. ... A strong adsorption capacity was needed to facilitate the decomposition of H₂O₂ into •OH radicals during the reaction-triggering step (Supplementary Fig. 29a), while a weak adsorption capacity was favorable for the breaking of the M–O bond during the conversion step (Supplementary Fig. 29b). Therefore, a volcano-type relationship between χ and M–O binding strength was established by the trade-off effects (Fig. 5f)." I cannot be fully convinced by the assumption. Which is the rate-determining step of the reaction: the activation of C-H in CH₄ or the formation and/or triggering step of •OH? Why the strong adsorption for the formation of M-O bonds while the weak adsorption for the break of the M-O bond is required?

2. The author claimed that in the Pd 3d XPS spectra, there is no observed overlapping interference between Pd 3d and Au 4d. XPS is a surface analysis technique. Why are there no Au atoms on the nanosheet surface? Did the author consider other possible structures? In addition, are the models for DFT calculations correct?

Dear Reviewers and Editor,

Many thanks for your time and work on our manuscript. Based on your valuable comments and suggestions, we have revised the manuscript carefully by supplementing additional discussions. We appreciate your comments, which have greatly enriched this manuscript. Details about the revisions and our responses to the reviewers' comments are provided below in our point-by-point response.

Reviewers' comments:

Reviewer #1 (Remarks to the Author):

I have a few questions/edits before recommending for final publication.

1. What is a footstone?

Response: Thanks for your comments. We have provided a more accurate expression to replace the “footstone” word in the revised manuscript.

Revision in the revised manuscript (Page 3)

Methane (CH₄) is a promising, clean, and cost-effective feedstock for producing high-value chemicals, particularly methanol (CH₃OH), because of its versatility as an energy carrier and the platform molecule of fine chemicals, like olefins and aromatics, which is an important pillar of the chemical industry.

2. Does the catalyst deactivate after 30 minutes?

Response: Thanks for your insightful comments. The catalyst has not been deactivated after 30 minutes. As shown by Fig. 3e, the yield of CH₃OH increase gradually with increasing reaction time, but level off after 30 minutes. Meanwhile, the selectivity of CH₃OH is positively dependent on the yield of CH₃OH over 30 minutes, but it begins to decrease with the longer reaction time. This phenomenon indicates that the catalyst cannot catalyze the CH₄ oxidation to CH₃OH after 30 minutes. However, the recycle experiments prove the continuous catalytic activity of the catalyst over 10 cycles (Fig. 3b). Therefore, above results prove that the catalyst has not been deactivated after 30 minutes. And the reasonable guess is that the DOMM performance

is limited by the concentration of in-situ generated H_2O_2 , and lower H_2O_2 concentration after 30 minutes cannot provide enough $\bullet\text{OH}$ radical to activate the C-H bond of CH_4 molecules.

3. In Figure 3E, there is no temperature listed.

Response: Thanks for your valuable comments. The reaction temperature has been listed in the revised manuscript.

Revision in the revised manuscript (Page 26)

Fig. 3 | DOMM performance. **a** The catalytic performance of DOMM for Pd, Pd_xAu_y , and Au NS. **b** Reaction tests for the recycle and regeneration of the Pd_3Au_1 catalyst. **c** Comparison of catalytic performance and CH_3OH selectivity for CH_4 direct conversion with various catalysts. **d** The catalytic performance with different reaction temperatures for Pd_3Au_1 NS. **e** The catalytic performance with different reaction times for Pd_3Au_1 NS at 70 $^{\circ}\text{C}$. **f** The catalytic performance with different CH_4 vol. Pd_3Au_1 NS. All other conditions remain the same: 10 mL of water, 1 mg of catalyst, feed gas at 3.0 MPa with 1.1% H_2 /2.2% O_2 /67.2% CH_4 /20.57% Ar/8.93 % He. Each reaction was tested three times to obtain the error bars.

Reviewer #2 (Remarks to the Author):

The comments have been mainly addressed, however, a few are still unclear.

Point 2: it was recommended to reword footstone however I see that "perhaps footstone" does not make sense.

It would be better to remove the word "perhaps" from the text. This is a personal phrasing and the authors should word as they see best.

Response: Thanks for your valuable comments. We have provided a more accurate expression to replace the “footstone” word and removed the word “perhaps” in the revised manuscript.

Revision in the revised manuscript (Page 3)

Methane (CH₄) is a promising, clean, and cost-effective feedstock for producing high-value chemicals, particularly methanol (CH₃OH), because of its versatility as an energy carrier and the platform molecule of fine chemicals, like olefins and aromatics, which is an important pillar of the chemical industry.

point 10: "mapping..." was not the correction rather "shown" changed to "show" which has been done. mapping "images" should be reinserted.

Response: Thanks for your comment. The word “mapping show an” has been corrected to “mapping images show an”.

Revision in the revised manuscript (Page 5)

The high-angle annular dark field (HAADF) STEM images and corresponding atomically resolved elemental mapping images show an ordered atomic arrangement and a regular geometry profile (Fig. 1i).

Point 13: The query was concerned with adding the details of the ICP method which I am not sure is included in the experimental section.

Response: Thanks for your valuable comment. Details of the ICP method have been added in the supplementary information.

Revision in the revised supplementary information (Page 3)

Supplementary Fig. 2 Standard curves of (a) Pd and (b) Au concentration with the corresponding signal values of the ICP-OES.

Revision in the revised supplementary information (Page 35)

Supplementary Table 2. The Pd/Au atomic ratios of Pd_xAu_y NS from the ICP-OES measure.

Catalysts	m_0 (g)	V_0 (mL)	Elements	C_0 (mg/L)	Dilution factor: f	C_1 (mg/L) ^ψ	C_x (mg/kg) ^ξ	Pd/Au molar ratio ^Φ
Pd ₃₃ Au ₁	0.0146	10	Au	0.60	100	60	41095.89	33:1
			Pd	10.77		1077	737575.34	
Pd ₆ Au ₁	0.0297	10	Au	4.97	100	497	167426.60	6:1
			Pd	16.36		1636	550872.05	
Pd ₃ Au ₁	0.0273	10	Au	5.61	100	561	205494.51	3:1
			Pd	9.19		919	336634.07	
Pd ₁ Au ₁	0.0305	10	Au	17.31	100	1731	567868.85	1:1
			Pd	10.07		1007	330131.15	

m_0 is weighing by analytical balance.

V_0 is the volume of the fixed volume after dissolution.

C_0 is calculated by the external standard method.

$$\psi: C_1 = C_0(\text{mg/L}) \times f.$$

$$\xi: C_x(\text{mg/kg}) = \frac{C_1(\text{mg/L}) \times V_0(\text{mL}) \times 10^{-3}}{m(\text{g}) \times 10^{-3}}$$

$$\Phi: \text{Pd/Au molar ratio} = \frac{n_{\text{Pd}}(\text{mol})}{n_{\text{Au}}(\text{mol})} = \frac{C_{x,\text{Pd}}(\frac{\text{g}}{\text{kg}}) \times m_0(\text{kg}) \times 10^{-6}}{M(\text{Pd}, \frac{\text{g}}{\text{mol}})} / \frac{C_{x,\text{Au}}(\frac{\text{g}}{\text{kg}}) \times m_0(\text{kg}) \times 10^{-6}}{M(\text{Au}, \frac{\text{g}}{\text{mol}})}$$

C_0 represents the concentration of Pd or Au element of Pd_xAu_y NS with known amount in aqua regia.

This value was calculated through the following method. Firstly, the standard curves are drawn with the known Pd or Au concentration by the inductively coupled plasma optical emission spectroscopy (ICP-OES) (Supplementary Fig. 2). Next, Pd_xAu_y NS with known amount dissolved in aqua regia of 10 mL was tested to obtain the signals values of Pd and Au elements by using ICP-OES. Finally, C_0 of Pd or Au element of Pd_xAu_y NS with a known amount was calculated according to the standard curves.

Point 15: The query was asking for the values of the Pd area to be included so allow the calculation of the TOF to be followed. I am still unsure where the areas of Pd are shown and how these surface areas were obtained. See also point 31 below.

Response: Thanks for your insightful comments. We have provided the detailed calculation process of TOF, especially the values of the Pd area, in the revised manuscript and supplementary information.

Revision in the revised manuscript (Page 15)

TOF calculation

The TOF numbers were calculated based following equation:

$$TOF = \frac{n_{CH_3OH}}{n_{surface} \times T} = \frac{n_{CH_3OH}}{n_{metal} \times T \times \delta} = \frac{n_{CH_3OH}}{\frac{m_{cat} \times W}{M} \times T \times \delta} \quad (3)$$

Where n_{CH_3OH} is the mole of produced CH₃OH, $n_{surface}$ is the mole of surface metal atoms of Pd NS or Pd_xAu_y NS. n_{metal} is the mole of total metal atoms of Pd NS or Pd_xAu_y NS. m_{cat} is the mass of the catalyst including metal and support. W is the mass loading of the Pd NS or Pd_xAu_y NS, which is measured by using ICP-OES measurement. M is the atomic mass of Pd. T is the reaction time. δ is the molar percentage of surface Pd atom of Pd NS or Pd_xAu_y NS.

Calculation of the molar percentage of surface Pd atom (δ)

$$\delta = \frac{n_i}{n_j} = \frac{N \times n_{single Pd atom}}{n_j} = \frac{\frac{S}{S_{single Pd atom}} \times \frac{M}{N_A}}{n_j} \quad (4)$$

$$S = \frac{3\sqrt{3}}{2} \times t^2 + 6 \times t \times h \quad (5)$$

For one Pd NS, n_i is the amount of substance of surface Pd atom, and n_j is the amount of substance of total Pd atom. N is the number of surface atoms. S is the one Pd NS surface area. $S_{single Pd atom}$ is single Pd atom surface area. The density, volume, edge length and thickness are ρ , V , t , and h . Where $\rho=12.02 \text{ g/cm}^3$, $t=3 \times 10^{-6} \text{ cm}$, $h=1.5 \times 10^{-7} \text{ cm}$, $V=2.6 \times t^2 \times h=3.51 \times 10^{-18} \text{ cm}^3$, $S=2.61 \times 10^{-11} \text{ cm}^2$, $S_{single Pd atom}=1.3 \times 10^{-15} \text{ cm}^2$ (*Nat. Commun.* 2021, 12, 1218), $M=106.4 \text{ g/mol}$, $N_A=6.02 \times 10^{23} \text{ mol}^{-1}$, $n_j=(\rho \times V)/M=4.2 \times 10^{-17} \text{ mol}$. Thus δ_{Pd} of Pd NS was calculated as 8.4%. For $\delta_{Pd_xAu_y}=m_{Pd_x} \times \delta$, m_{Pd_x} is the atomic percentage of Pd in Pd_xAu_y NS.

Revision in the revised supplementary information (Page 41)

Supplementary Table 8. The turnover frequencies (TOF) of the Pd and Pd_xAu_y NS for DOMM.

Catalyst	$n_{CH_3OH} \text{ (mol)}$	$n_{surface} \text{ (mol)}$	$\delta \text{ (%)}$	$m_{cat.} \text{ (g)}$	$M \text{ (g/mol)}$	$W \text{ (%)}$	$T \text{ (h)}$	TOF (h^{-1})
Pd	3.64×10^{-6}	9.4×10^{-7}	8.4					92.2
$Pd_{33}Au_1$	4.89×10^{-6}	8.9×10^{-7}	8.0					139.8
Pd_6Au_1	5.66×10^{-6}	7.2×10^{-7}	7.2	1×10^{-3}	106.42	10	0.5	218.4
Pd_3Au_1	7.39×10^{-6}	5.8×10^{-7}	6.3					404.5
Pd_1Au_1	4.75×10^{-6}	3.3×10^{-7}	4.2					685.4

Point 17: should be "of the in-situ generated H_2O_2 ".

Response: Thanks for your comments. This phrase has been corrected in the revised manuscript.

Revision in the revised manuscript (Page 9)

Upon increasing the reaction time from 0.01 to 1.00 h, both the intensities of the O–H vibration peak and $*CH_3$ peak increased, indicating that the activation of the first C–H bond to form the adsorbed $*CH_3$ species was accomplished by the formation of $*OH$ derived from the dissociation of the in-situ generated of H_2O_2 .

Point 31: It is still unclear how the number of active sites has been determined. The amount of Pd atoms is relevant for productivity but TOF conventionally used surface sites, please comment on the number of Pd sites which are on the surface. This relates to point 15 above

Response: Thanks for your professional comments. As suggested in point 15, TOF calculation is related with the values of the Pd area or the number of surface Pd atoms. Therefore, TOF calculation of Pd NS and Pd_xAu_y

NS in this work is adopted by the amount of substance of surface Pd atoms, rather than the whole Pd atoms. According to equations (4) and (5), the amount of substance of surface Pd atoms is calculated through the hexagon Pd NS or Pd_xAu_y NS with a thickness of 1.6 nm. And the detailed calculation process had been listed in Point 15.

Reviewer #3 (Remarks to the Author):

The authors have added some new discussion in this revised manuscript to address my comments. However, I still have concerns for the reaction kinetics and the structure characterization of AuPd nanosheets.

1. The authors claimed the existence of a volcano-type relationship between M-O binding strength and catalytic performance. “During the DOMM process, methyl and hydroxyl groups were chemisorbed on the catalyst surface to form M-C and M-O bonds (M represents metal), respectively. ... A strong adsorption capacity was needed to facilitate the decomposition of H₂O₂ into •OH radicals during the reaction-triggering step (Supplementary Fig. 29a), while a weak adsorption capacity was favorable for the breaking of the M-O bond during the conversion step (Supplementary Fig. 29b). Therefore, a volcano-type relationship between χ and M-O binding strength was established by the trade-off effects (Fig. 5f).” I cannot be fully convinced by the assumption. Which is the rate-determining step of the reaction: the activation of C-H in CH₄ or the formation and/or triggering step of •OH? Why the strong adsorption for the formation of M-O bonds while the weak adsorption for the break of the M-O bond is required?

Response: Thanks for your valuable comment. According to Luo’s work (*Nat. Commun.* 2021, 12, 1, 1218), and both the energy barriers calculation and experimental results in our work, the CH₃OH formation can be regarded as the rate-determining step.

Previous literatures have demonstrated the radical mechanism of DOMM that •OH radical derived from H₂O₂ could activate the C-H bond of CH₄ molecule to form •CH₃ radical. However, little work has been done to reveal the role of catalyst for the formation of •OH radical. Our experimental results and mechanistic investigations emphasize that not only the reaction-conversion step, involving the activation of C-H bond and

the formation of CH₃OH, have a significant effect on the DOMM, but the reaction-triggering step, involving the decomposition of H₂O₂ to produce OH species, also has a non-negligible effect on the DOMM. As the volcano-type structure-performance relationship observed in the experimental results, both the reaction-triggering step and the reaction-conversion step should be considered together to obtain a complete depiction of the DOMM process.

A strong adsorption capacity is needed to facilitate the decomposition of H₂O₂ into •OH radicals during the reaction-triggering step (Supplementary Fig. 30a), while a weak adsorption capacity is favorable for the breaking of the M–O bond during the conversion step (Supplementary Fig. 30b). However, for catalysts that bind OH species too strong, the performance of DOMM is limited by the slow step of CH₄ conversion to CH₃OH. On the contrast, for catalysts that bind OH species too weak, the performance is limited by insufficient •OH radicals. Thus, a moderate adsorption capacity of OH species is needed for the effective DOMM process. Therefore, a volcano-type relationship between χ and M–O binding strength was established by the trade-off effects (Fig. 5f).

2. The author claimed that in the Pd 3d XPS spectra, there is no observed overlapping interference between Pd 3d and Au 4d. XPS is a surface analysis technique. Why are there no Au atoms on the nanosheet surface? Did the author consider other possible structures? In addition, are the models for DFT calculations correct?

Response: Thanks for your insightful comments. The structural characterizations, including atomic-resolution AC-STEM and XAS, have clearly demonstrated the existence of Au atoms on the Pd NS surface. However, the absence of the Au 4d signal peaks in XPS spectra is due to the low content of Au atoms in Pd_xAu_y NS (**Table R1**). Moreover, the Au 4f orbital is frequently analyzed as the characteristic peak due to its stronger signal intensity, and Au 4d orbital with weaker signal intensity is hard to detect. This phenomenon has also been proven by other reported literature, which also didn't show the signal peaks of Au 4d orbital (*Nat. Catal.* 2021, 4, 575–585; *Sci. Adv.* 2016, 2, e1600858; *Adv. Mater.* 2017, 29, 1701331; *J. Am. Chem. Soc.* 2019, 141,

4791–4794). Therefore, ultrathin Pd_xAu_y NS with Au atoms located on the surface has been successfully prepared, and DFT models with Au atoms dispersed on the surface are adopted for mechanistic explorations.

Table R1. The Pd and Au atomic % of Pd_xAu_y NS supported on carbon blacks from the XPS measure.

Catalysts	Atomic content of C (%)	Atomic content of Pd (%)	Atomic content of Au (%)	Pd/Au atom ratio
Pd ₃₃ Au ₁	99.06	0.91	0.03	30
Pd ₆ Au ₁	99.19	0.83	0.13	6
Pd ₃ Au ₁	99.02	0.79	0.19	4
Pd ₁ Au ₁	99.00	0.71	0.29	2

Finally, we again appreciate the editor and the reviewers for your valuable time and insightful comments, which were extremely helpful in improving the quality of our paper and enhancing our future work. We also hope that Reviewers will appreciate our great efforts put into the revision, and this version will be deemed acceptable for publication in *Nature Communications*.

REVIEWERS' COMMENTS

Reviewer #1 (Remarks to the Author):

With the most recent revisions, I recommend this manuscript for publication in Nature Communications.

Reviewer #3 (Remarks to the Author):

The authors have addressed all my comments in a convincing way. The paper is now well-rounded and I would recommend the manuscript for its publication in Nature Communications.

Dear Reviewers and Editor,

Many thanks for your time and work on our manuscript. Based on your valuable comments and suggestions, we have revised the manuscript carefully by supplementing additional discussions. We appreciate your comments, which have greatly enriched this manuscript. Details about our responses to the reviewers' comments are provided below in our point-by-point response.

Reviewers' comments:

Reviewer #1 (Remarks to the Author):

With the most recent revisions, I recommend this manuscript for publication in Nature Communications.

Response: We extend our heartfelt gratitude for the time and effort dedicated to our manuscript.

Reviewer #3 (Remarks to the Author):

The authors have addressed all my comments in a convincing way. The paper is now well-rounded and I would recommend the manuscript for its publication in Nature Communications.

Response: We would like to express our deepest appreciation for the time and effort devoted to reviewing our manuscript.

Finally, we again appreciate the editor and the reviewers for your valuable time and insightful comments, which were extremely helpful in improving the quality of our paper and enhancing our future work.